# Hyperacidification of *Citrus* fruits by a vacuolar proton-pumping P-ATPase complex

Pamela Strazzer [1], Cornelis E. Spelt[1], Shuangjiang Li [1], Mattijs Bliek[1], Claire T. Federici[2], Mikeal L. Roose[2], Ronald Koes [1] & Francesca M. Quattrocchio [1]

The sour taste of *Citrus* fruits is due to the extreme acidification of vacuoles in juice vesicle cells via a mechanism that remained elusive. Genetic analysis in petunia identified two vacuolar P-ATPases, PH1 and PH5, which determine flower color by hyperacidifying petal cell vacuoles. Here we show that *Citrus* homologs, CitPH1 and CitPH5, are expressed in sour lemon, orange, pummelo and rangpur lime fruits, while their expression is strongly reduced in sweet-tasting "acidless" varieties. Down-regulation of *CitPH1* and *CitPH5* is associated with mutations that disrupt expression of MYB, HLH and/or WRKY transcription factors homologous to those activating *PH1* and *PH5* in petunia. These findings address a long-standing enigma in cell biology and provide targets to engineer or select for taste in *Citrus* and other fruits.

[1] Swammerdam Institute for Life Sciences, University of Amsterdam, Science Park 904, 1098 XH Amsterdam, The Netherlands. [2] Department of Botany and Plant Sciences, University of California, Riverside, CA 92521, USA. These authors contributed equally: Ronald Koes, Francesca Quattrocchio. Correspondence and requests for materials should be addressed to R.K. (email: r.e.koes@uva.nl) or to F.M.Q. (email: f.quattrocchio@uva.nl)

Citrus are widely used for the consumption of fruit flesh and juices. Modern citrus varieties were generated over thousands of years by intraspecific and interspecific crosses of a handful of species combined with clonal propagation[1–3]. Acidity is a major trait determining the taste and use of citrus fruits and selection by breeders and producers has generated a broad palette of sour and "sweet" (i.e., non-sour) varieties of lemons, oranges, pummelos, and other citrus fruits[4]

For a sour taste, food/liquid should have (i) a high concentration of free $H^+$ ions (low pH), which is sensed by acid-sensitive cells in taste buds, presumably via an $H^+$-selective channel[5] and (ii) a certain pH-buffering capacity to prevent the liquid from being neutralized by the saliva. The acidity (low pH) of *Citrus* fruits is determined by the pH of the vacuoles in juice vesicle cells, which can be as low as 2 in sour lemons and lime[6–8]. The steep proton gradient across the vacuolar membrane (tonoplast) drives massive transport of citrate into the vacuole via a mechanism that is only partially understood[9,10]. As citrate enters the vacuole in dissociated form (citrate$^{3-}$), it increases its buffer capacity, which contributes to the sour taste[11], but does not lower the pH. How juice vesicle cells can hyperacidify their vacuoles to such an extreme extent remained elusive in spite of extensive biochemical work[8,9,12–15].

In most plant cells, the cytoplasm is about neutral and the vacuolar lumen mildly acidic. The (moderate) pH gradient across the tonoplast is generated by vacuolar-ATPases (V-ATPases)[16,17], which are complex multi-subunit proton pumps found in both animals and plants[18,19]. V-ATPases translocate 2–4 protons per hydrolyzed ATP ($H^+$/ATP = 2–4) depending on the pH on both sides of the membrane[20,21] and may in theory acidify vacuoles down to pH ≈ 3.5, when operating without kinetic inhibition, which it rarely if ever occurs in vivo, and in its "lowest gear" ($H^+$/ATP = 2). Further acidification to pH < 3, as in lemon vacuoles, would require $H^+$/ATP ratios <2 (ref. [20]). Acid lemons indeed contain a proton-pump activity with $H^+$/ATP = 1 stoichiometry[8,12,14,15]. However, the nature of this fruit-specific proton pump has remained elusive ever since, because it could not be completely purified[13,15].

In petunia, mutations in one of the seven *PH* loci (*PH1–PH7*) reduce the acidity of petal vacuoles and petal homogenates, resulting in a blue flower color[22,23]. *PH3* and *PH4* encode transcription factors of the WRKY and MYB family, respectively, which together with the helix–loop–helix protein ANTHOCYANIN1 (AN1) and the WD-repeat protein AN11, form a complex (WMBW) that activates genes involved in vacuolar acidification[24–27]. *PH1* and *PH5* are the major downstream genes involved in vacuolar hyperacidification[28]. *PH5* encodes a $P_{3A}$-ATPase proton pump that resides in the tonoplast instead of the plasma membrane where other $P_{3A}$-ATPases reside[29,30]. PH1 is a $P_{3B}$-ATPase similar to bacterial Mg transporters and also resides in the tonoplast. It has no known transport activity on its own but can bind to PH5 and promote PH5 proton-pumping activity[28] and has an additional role in membrane/protein trafficking to the vacuole[31].

PH1 and PH5 homologs are widely conserved among flowering plants, including species without colored petals, suggesting that their function is not confined to flower pigmentation[30,32]. Since *CitPH5* homologs are expressed in sour lemons and oranges[33–35] and P-ATPase proton pumps can theoretically generate steeper proton gradients (because $H^+$/ATP = 1), we investigated whether the PH5/PH1 complex might be the proton pump that acidifies *Citrus* vacuoles to such extreme extent.

Analyzing a collection of lemon, orange, and pummelo varieties (Supplementary Table 1), we found that *PH1* and *PH5* homologs are highly expressed in all acidic (low pH) fruits but are downregulated in non-acidic (high pH) fruits, due to inactivating

mutations in CitAN1 (sweet lemon and sweet oranges) or regulatory mutations that inactivate *CitAN1*, *CitPH3*, and/or *CitPH4* expression.

## Results

### *CitPH1* and *CitPH5* expression in Faris lemons.

Lemon trees of the variety 'Faris' produce branches bearing either sour or sweet (non sour) fruits, enabling a comparison of sweet and sour fruits grown in the same conditions[33]. However, sweet and sour fruits are not necessarily isogenic, because 'Faris' is a graft chimera in which the L1 tunica layer of the shoot meristem derives from an unknown variety related to 'Millsweet' limetta and the L2 layer from a standard sour lemon[33]. Branches of 'Faris' trees carrying sour fruits (Fso) have purple immature leaves, a purple blush on the lower petal epidermis, and dark spots at the chalazal end of the seeds, whereas the branches bearing sweet fruits (Fsw) lack purple pigmentation on leaves, petals, and seeds (Fig. 1a). The juice of Fsw fruits is less acidic (pH 5.1) than that of Fso or 'Frost Lisbon' (Fli) (pH 2.5), a standard lemon with strong sour taste, and Fsw fruits contain less titratable acid than Fso or Fli fruits (Fig. 1b). There was, however, no correlation between soluble solid content (Brix) and juice taste (Fig. 1b), indicating that juice pH and/or titratable acid rather than the sugars, which are major components of total soluble solids, determine the different taste of Fsw and Fso fruits.

In melon (*Cucumis melo*) and tomato (*Solanum lycopersicum*), which both lack *PH1* and *PH5* homologs[30], the (mild) acidification of the fruit flesh requires a membrane transporter of unknown function, known in melon as PH or SOUR[36]. We identified the *SOUR* homolog of *Citrus* (*CitSO*) and found that *CitSO* mRNAs in Fsw and Fli fruits are expressed at similar levels and lack mutations with an obvious effect on the expression and/or activity of the encoded protein, indicating that *CitSO* does not contribute to the differences in their acidity (Supplementary Figs. 1 and 2).

Analysis of the *Citrus limon* homologs *CitPH1* and *CitPH5* identified previously[30] revealed that both are expressed in low-pH Fli and Fso lemons but not in high-pH Fsw lemons (Fig. 1c, Supplementary Fig. 3a). Expression of the *Citrus* homolog of *MAC9F1*, a petunia gene of unknown function that is activated by the same transcription factors as *PH1* and *PH5*[37], is also downregulated in Fsw juice vesicles similar to *CitPH1* and *CitPH5* (Fig. 1c).

These findings indicate that (i) the CitPH1-PH5 heterodimeric proton pump is involved in the hyperacidification of Fso and Fli fruits and (ii) that the reduced acidity of Fsw fruits results from a mutation in a regulatory gene controlling the expression of *CitPH1*, *CitPH5*, and other genes like *CitMAC9F1*.

### Inactivation of *CitAN1* causes the reduced acidity in Fsw.

To identify the mutation(s) responsible for the reduced expression of *CitPH1* and *CitPH5* in Fsw fruits, we identified *Citrus* homologs of the transcription regulators *AN1*, *AN11*, *PH4*, and *PH3* (Supplementary Fig. 4), which drive *PH1* and *PH5* expression in petunia petals and seeds[28,29]. Public RNA-seq data[38] indicate that in the *Citrus sinensis* (sweet orange) variety 'Valencia' *CitPH4* is predominantly expressed in fruits, like *CitPH5* and *CitMAC9F1*, whereas *CitAN1* is also expressed in flowers, where it most likely drives anthocyanin synthesis in concert with other MYB proteins (Supplementary Fig. 5).

Reverse transcriptase–polymerase chain reaction (RT-PCR) experiments using primers complementary to the 5′ and 3′ end of the *CitAN1* coding sequence revealed that Fli and Fso juice vesicles express full-size *CitAN1* mRNAs (Fig. 1d; Supplementary Fig. 3b), which encode a functional protein (see below). Both

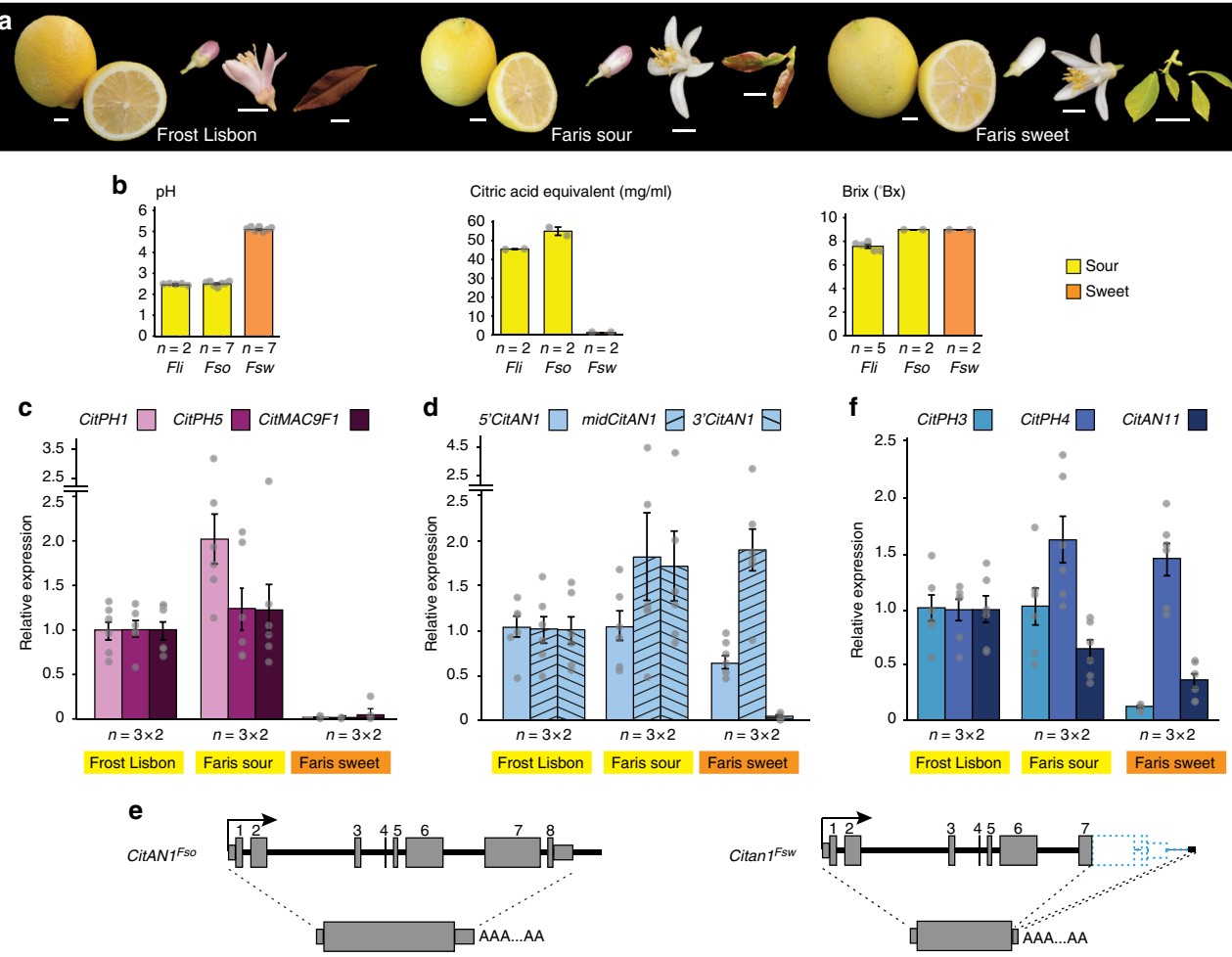

**Fig. 1** Analysis of 'Frost Lisbon' (*Fli*), 'Faris' sour (*Fso*), and 'Faris' sweet (*Fsw*) lemons. **a** Phenotypes of *Fli*, *Fso*, and *Fsw* fruits, flowers, and leaves. Scale bars represent 1 cm. **b** pH, titratable acid, and soluble solid content (°Brix) in juice from (nearly) mature *Fli*, *Fso*, and *Fsw* fruits (mean ± SE). **c** Real-time RT-PCR of *CitPH1*, *CitPH5*, and *CitMAC9F1* mRNA in juice vesicles from (nearly) mature *Fli*, *Fso*, and *Fsw* fruits. **d** Real-time RT-PCR of *CitAN1* mRNA in *Fli*, *Fso* and *Fsw* juice cells, using primers that amplify the 5′, middle, and 3′ part of the mRNA. **e** Structure of the *CitAN1* alleles and their transcripts in *Fso* and *Fsw* juice vesicles. **f** Real-time RT-PCR of *CitPH3*, *CitPH4*, and *CitAN11* mRNA in juice vesicles from (nearly) mature *Fli*, *Fso*, and *Fsw* fruits. Values in **c**, **d**, **f** are mean ± SE; *n* = number of samples from different fruits × number of technical replicates of each. Source data are provided as a Source Data file

*CitAN1* alleles of *Fso* juice vesicles consist of seven exons, of which one (exon 4) is skipped in a fraction of the mature *CitAN1* mRNA (Fig. 1e; Supplementary Fig. 3), and are distinguishable by sequence polymorphisms in *CitAN1* (single-nucleotide polymorphism (SNPs) and a 1661-bp transposon insertion in the 5′ flanking region) and in the genes immediately upstream (*CitFAR-like*) and downstream (*CitTFIIH-like*) (Supplementary Figs. 6–8).

*Fsw* fruits, by contrast, lack full-size *CitAN1* mRNAs and instead express truncated *CitAN1* transcripts, which span the 5′ and middle region of the mRNA but lack the 3′ part (Fig. 1d, e and Supplementary Fig. 3). Analysis of genomic and 3′-RACE cDNA fragments revealed that the *citan1Fsw* allele(s) contain an identical 1.3-kb deletion with breakpoints in exon 7 and 143 bp downstream the normal polyadenylation site, resulting in a transcript that lacks exon 8 and most of exon 7 to terminate at a cryptic polyadenylation site 57 bp downstream from the deletion breakpoint (Fig. 1e and Supplementary Fig. 9). Sequencing of PCR products from *CitAN1* and the flanking genes *CitFAR-like* and *CitTFIIH-like* revealed no sequence polymorphism (Supplementary Fig. 8). This indicates that in *Fsw* fruits the genomic *CitAN1* region is either homozygous or hemizygous over

a larger deletion in the sister chromosome that spans *CitAN1*, *CitFAR-like*, and *CitTFIIH-like*.

Transcripts of the *PH4* homolog *CitPH4* accumulate at similar levels in *Fso* and *Fsw* fruits (Fig. 1f), excluding that the high-pH phenotype results from downregulation of *CitPH4*. *Fso* and *Fsw* fruit both contain two distinct *CitPH4* alleles distinguishable by few SNPs and triplet repeats of variable length, resulting in several amino acid replacements and polyglutamine (Q) tracks of variable length, which may affect protein function and/or stability (Supplementary Figs. 10–11). The sequences of *CitAN11Fso* and *CitAN11Fsw* did not reveal differences with an obvious negative effect on *Fsw* protein activity (Supplementary Fig. 12), and in fruits, mRNA expression levels of the two alleles are comparable (Fig. 1e).

The finding that juice vesicles of *Fso* and *Fsw* fruits contain distinct alleles for multiple genes, including *CitAN1*, *CitPH4*, *CitAN11*, *CitFAR-like*, and *CitTFIID-like*, suggests that the occurrence of sweet and sour fruits results from atypical periclinal cell division(s) by which a daughter of a cell in the L2 meristem layer, containing the genome of a standard sour lemon, invaded L1 and displaced the mutant 'Millsweet'-like L1 cells, or vice versa, as in other chimeras[39,40], rather than the somatic reversion of an unstable (epi)allele.

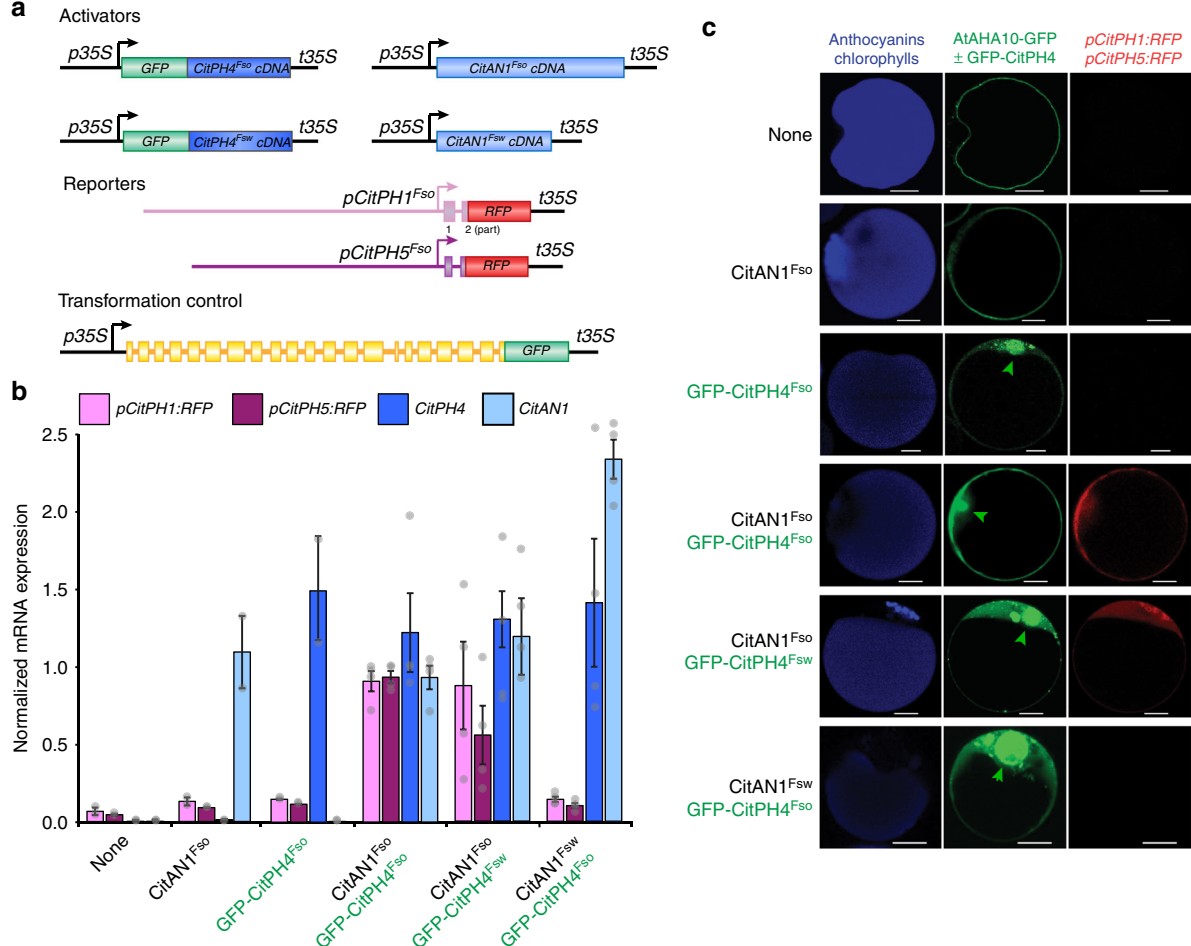

**Fig. 2** Activation of *CitPH1* and *CitPH5* promoters by CitAN1 and CitPH4. **a** Structure of effector, reporter, and control genes used for transient transformation of protoplast from the epidermis of petunia *ph4* petals. **b** Relative expression (real-time RT-PCR of effector (*CitAN1* and *CitPH4*) and reporter (*pCitPH1:RFP* and *pCitPH5:RFP*) mRNA expression in petal protoplasts transformed with different effectors [mean ± SE from 2 technical PCR replicates of 4 independent transformation experiments (*n* = 4 × 2)]. Expression of the co-transformed *35S:AHA10-GFP* gene was used for normalization. **c** Confocal micrographs of petunia *ph4* epidermal petal protoplasts, with anthocyanins in the central vacuole, from stage 6 flowers, 24 h after transformation of *35S:AHA10-GFP*, the reporters *pCitPH1:RFP*, *pCitPH5:RFP*, and different combinations of the effectors *35S:GFP-CitPH4Fso*, *35S:GFP-CitPH4Fsw*, *35S:CitAN1Fso*, and *35S:CitAN1Fsw*. Note that GFP-CitPH4 expression is visible in the nucleus (green arrows). Images shown are representative of tens to hundreds of cells observed. Size bars equal 10 μm. Source data are provided as a Source Data file

To examine whether *CitPH1* and *CitPH5* are transcriptionally activated by *CitAN1* and *CitPH4* and whether the deletion in *CitAN1* and/or the polymorphisms in *CitPH4* coding sequence impair the transcription of *CitPH1* and *CitPH5* in *Fsw* lemons, we performed transient expression in protoplasts from *ph4* petunia petals, where endogenous *PH1* and *PH5* promoters are inactive. Therefore, we expressed various combinations of CitAN1Fso, CitAN1Fsw, and green fluorescent protein (GFP) fusions of CitPH4Fso or CitPH4Fsw from the constitutive 35S promoter and measured the expression of red fluorescent protein (RFP) reporter genes that were translationally fused to the *CitPH1Fso* or *CitPH5Fso* promoters (Fig. 2a). To identify transformed cells and normalize the expression levels for transformation efficiency, we co-transformed *p35S:AHA10-GFP*, which encodes a GFP fusion of the *Arabidopsis* PH5-homolog AHA10 that localizes in tonoplast of the central vacuole[32], and the nuclear localized GFP-CitPH4.

Expression of either CitAN1 or GFP-CitPH4 alone was insufficient to induce the *CitPH1* or *CitPH5* reporters in protoplasts (Fig. 2b, c; Supplementary Fig. 13). However, co-

expression of CitAN1 and GFP-CitPH4 strongly induced *pCitPH1:RFP* and *pCitPH5:RFP* expression (Fig. 2b) in most if not all cells expressing GFP-CitPH4 and AHA10-GFP (Fig. 2c) and was independent from other (petunia) regulators of the anthocyanin/pH pathway as it also occurred in white mesophyll cells (Supplementary Fig. 13) where endogenous *AN* and *PH* genes are not expressed[26,27,29]. The *CitPH4FSo* and *CitPH4Fsw* alleles activated the *CitPH1* and *CitPH5* reporters with similar efficiency, whereas the truncated AN1 protein encoded by *an1Fsw* proved unable to induce *pCitPH1:RFP* or *pCitPH5:RFP* expression.

These findings indicate that the mutation in *citan1Fsw* is responsible for the reduced *CitPH1* and *CitPH5* expression in the *Fsw* fruits and very likely also the loss of anthocyanins in leaves and flowers on the branches bearing *Fsw* fruits, whereas the polymorphisms in the *CitPH4Fsw* and *CitPH4Fso* have little or no effect. The reduced expression of *CitPH3* in *Fsw* juice cells (Fig. 1f) is likely due to the *citan1Fsw* mutation, since in petunia petals *PH3* is (partially) regulated by AN1 and PH4 and essential for transcription of *PH1* and *PH5*[27].

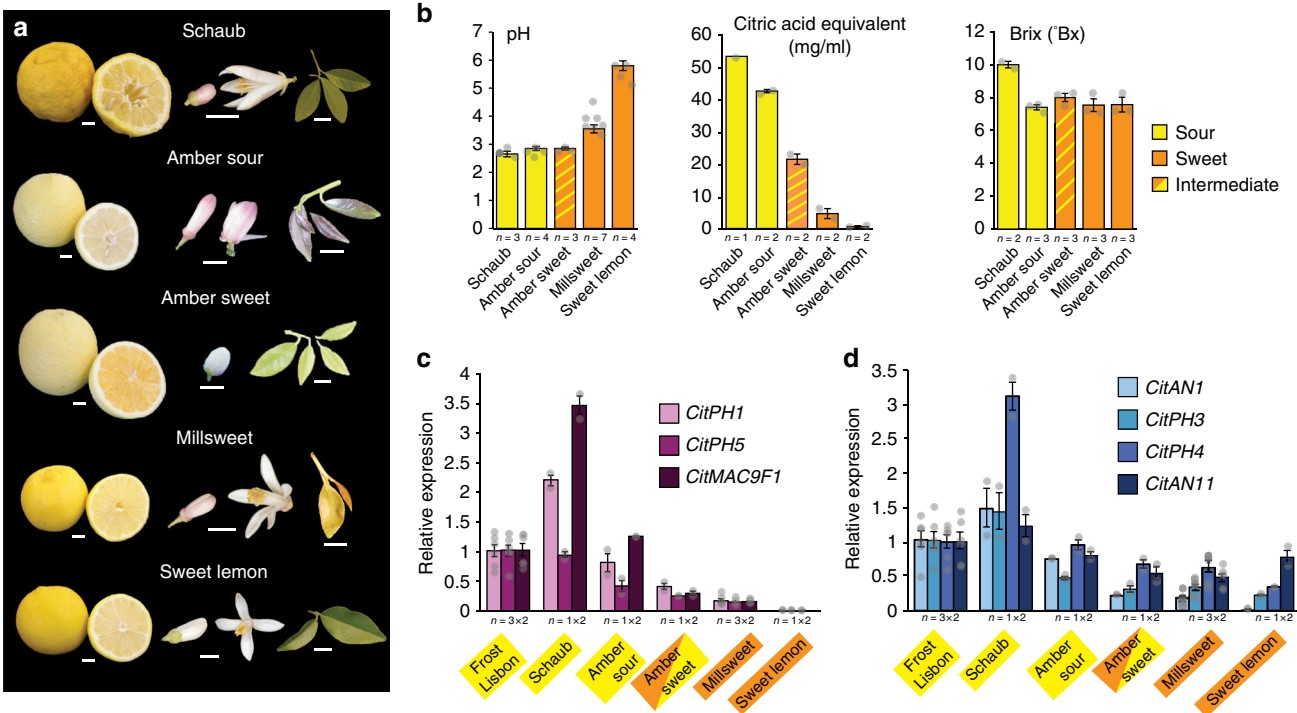

**Fig. 3** Analysis of 'Schaub', 'Amber', 'Millsweet', and Sweet lemon varieties. **a** Fruits, flower, and leaf phenotypes of the analyzed varieties. Size bars represent 1 cm. **b** pH, titratable acid, and soluble solid content (°Brix) in juice from different varieties (mean ± SE). **c** Real-time RT-PCR of *CitPH1*, *CitPH5*, and *CitMAC9F1* mRNA juice vesicles. **d** Real-time RT-PCR of *CitAN1*, *CitPH3*, *CitPH4*, and *CitAN11* mRNA in juice vesicles. Values in **c**, **d** are mean ± SE; *n* = number samples from different fruits × number of technical replicates of each. For **c**, **d**, full-size nearly mature fruits were analyzed. Relative mRNA expression levels from 'Frost Lisbon' were taken from Fig. 1 and are repeated here for comparison. Source data are provided as a Source Data file

**CitPH1 and CitPH5 expression in other sweet lemons.** To support that the reduced acidity of *Fsw* fruit was caused by mutation in *CitAN1* and consequent loss of *CitPH1* and *CitPH5* expression, rather than by independent mutations affecting unrelated pathway(s), we analyzed additional sweet and sour lemon varieties (Fig. 3a). 'Schaub' rough lemon is a non-edible citrus variety, unrelated to standard lemons and generally used as rootstock. 'Millsweet' limetta (*Citrus limetta*) and unnamed Sweet lemon (*Citrus limettioides*) bear sweet-tasting fruits, which have a similar soluble solid content (Brix) but have reduced juice acidity and titratable acid content. as well as *CitPH1*, *CitPH5*, and *Cit-MAC9F1* mRNA expression compared to 'Schaub' and 'Frost Lisbon' fruits (Fig. 3b, c).

*CitAN1*, *CitPH3*, and *CitPH4* transcripts are low in abundance in 'Millsweet' limetta and Sweet lemon compared to low-pH fruits, whereas *CitAN11* is expressed at similar levels as in low-pH fruits (Fig. 3d). 'Schaub', sweet 'Amber', and sour 'Amber' lemons have two *CitAN1* alleles with wild-type structure (Fig. 4), differing only for a few SNPs in *CitAN1* and the flanking genes *CitFAR-like* and *CitTFIIH-like* (Supplementary Figs. 8–9). 'Millsweet' instead contains one allele with a wild-type structure and a truncated allele (*citan1^{Msw}*) with a 1.3-kb deletion. The deletion in *citan1^{Msw}* has the same breakpoints as that in *citan1^{Fsw}* (Fig. 4, Supplementary Fig. 9) suggesting that both alleles originate from the same deletion event, which apparently occurred in the distant past as *citan1^{Fsw}* and *citan1^{Msw}* and the flanking *CitFAR-LIKE* and *CitTFIIH-like* genes acquired since then several polymorphisms (Supplementary Figs. 8–9). As the first plant containing this 3' deletion allele of *CitAN1* was most likely heterozygous, homozygotes appeared later in progenies segregating for the deletion allele from various crosses, because of which today's varieties can be homozygous (*Fsw*) or heterozygous (*Msw*) for this allele.

PCR analysis and partial sequencing showed that Sweet lemon contains a *CitAN1* allele of normal size and a truncated allele with a deletion in 3' end that may be similar to the deletion in *citan1^{Fsw}* and *citan1^{Msw}* (Supplementary Fig. 14). One of the Sweet lemon alleles, presumably the full-size allele, contains in its upstream region the same 1.7-kb transposon insertion as found in the 'Faris' sour and 'Frost Lisbon' alleles (Fig. 4; Supplementary Fig. 6). Given that, in 'Millsweet' limetta and Sweet lemon fruits mRNAs from both the full-size and a truncated *CitAN1* allele are downregulated, in spite of their very different origin, it is most likely that their reduced expression is caused by a mutation in an upstream *trans*-acting factor, rather than independent *cis*-acting mutations in each allele.

'Amber' is a chimera, like 'Faris', that originated as a variant tree in a grove of 'Eureka' lemons. Some branches on the 'Amber' tree bear pale green leaves, white flowers, and sweet-tasting fruits with "amber"-colored flesh, while other branches bear purplish young leaves, flowers with a purple blush, and fruits with yellow flesh and sour taste (Fig. 3a). Sweet 'Amber' lemons have a low juice pH similar to sour 'Amber', 'Schaub', and 'Frost Lisbon' lemons and express similar *CitPH1* and *CitPH5* mRNA levels as sour 'Amber' fruits and only slightly less than in 'Frost Lisbon'. Expression of the *CitAN1* allele(s) in 'Amber' sour fruits, which contain the same transposon insertion as *CitAN1^{Fso}*, and those in sweet 'Amber' fruit, which lack this insertion (Supplementary Fig. 6), differs less than two-fold, indicating that this transposon has little or no effect on the expression of *CitAN1* and downstream genes. Thus 'Amber' sweet is essentially a low-pH variety that owes its sweet taste not from increased juice pH, but from the reduced buffer capacity, as measured by the amount of titratable acid (Fig. 3b), which might result from genetic defects in distinct pathways affecting, for example, citrate transport into the vacuole.

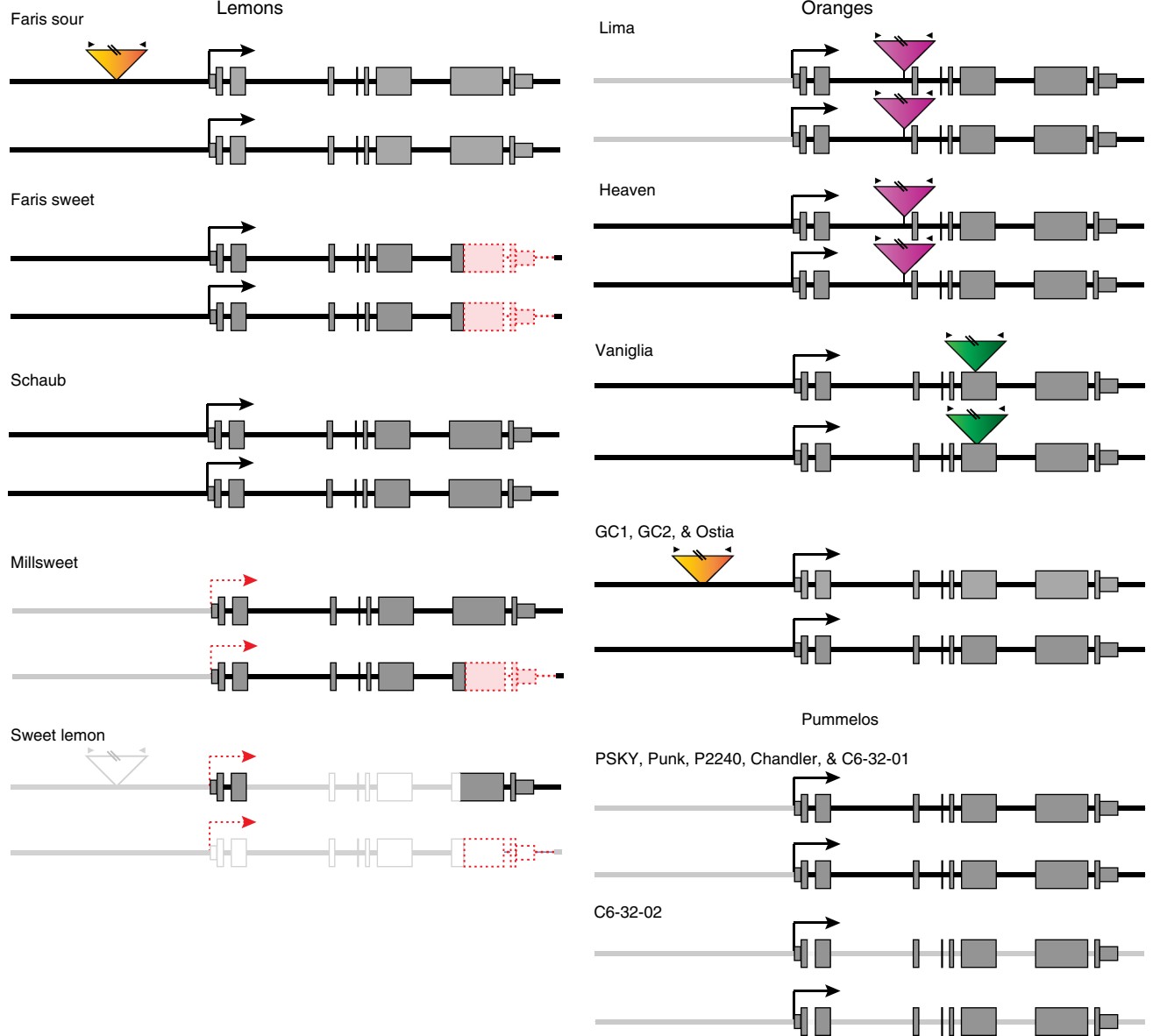

**Fig. 4** Structure and expression of *CitAN1* alleles in distinct lemon, orange, and pummelo varieties. Exons are indicated by rectangles with 5′ and 3′ untranslated regions at reduced height. Deleted sequences are indicated with dotted red lines and pink filling and transposon insertions by triangles with different colors. The hooked arrow marks the transcription start. The red dotted hooked arrows in 'Millsweet' and Sweet lemon indicate (inferred) reduced transcription caused by reduced activity of an upstream regulator(s). Gene structures confirmed by sequencing are depicted as dark gray lines (introns and flanking DNA), filled rectangles (exons), or filled triangles (transposon insertion). Gene structures inferred from polymerase chain reaction alone are marked by gray lines and white filling

Taken together, we found that all varieties with low-pH lemons ('Frost Lisbon', 'Schaub', sour 'Amber', sweet 'Amber') express relatively high levels of *CitPH1* and *CitPH5* mRNA, whereas in all varieties with strongly reduced acidity ('Faris sweet', 'Millsweet', Sweet lemon) *CitPH1* and *CitPH5* expression is reduced due to independent mutations that affect upstream transcription regulators. This strongly supports the view that *CitPH1* and *CitPH5* are essential for the hyperacidification of vacuoles in lemon juice vesicles and sour taste of the fruit.

**CitAN1 and CitPH4 are downregulated in high-pH oranges.** Next, we extended our investigation to orange varieties (Fig. 5a). Sweet (*Citrus sinensis*) and sour oranges (*Citrus aurantium*) are both hybrids originating from pummelo and mandarin[1,3], with tastes ranging from sharply sour to insipidly sweet. We analyzed 'Pineapple' orange, a variety with moderate acid and sweet-rich taste, three varieties with a bland (non-sour) taste ('Lima', 'Vaniglia', and 'Orange of Heaven'), and sour oranges from two trees growing on different locations on Gran Canaria Island (small and large fruits) and one from Ostia, Italy. 'Pineapple' orange juice vesicles express *CitPH1* and *CitPH5* and *CitMAC9F1* at similar levels as sour oranges, which correlates with the low pH of the fruit juice, whereas the titratable acid content is much lower than in sour oranges (Fig. 5b, c).

The juice of 'Lima', 'Vaniglia', and 'Orange of Heaven' oranges has higher pH and lower titratable acidity than juice from the sour varieties from Ostia, Gran Canaria1 (*GC1*) and *GC2*, whereas the soluble solid content (Brix) is similar and therefore

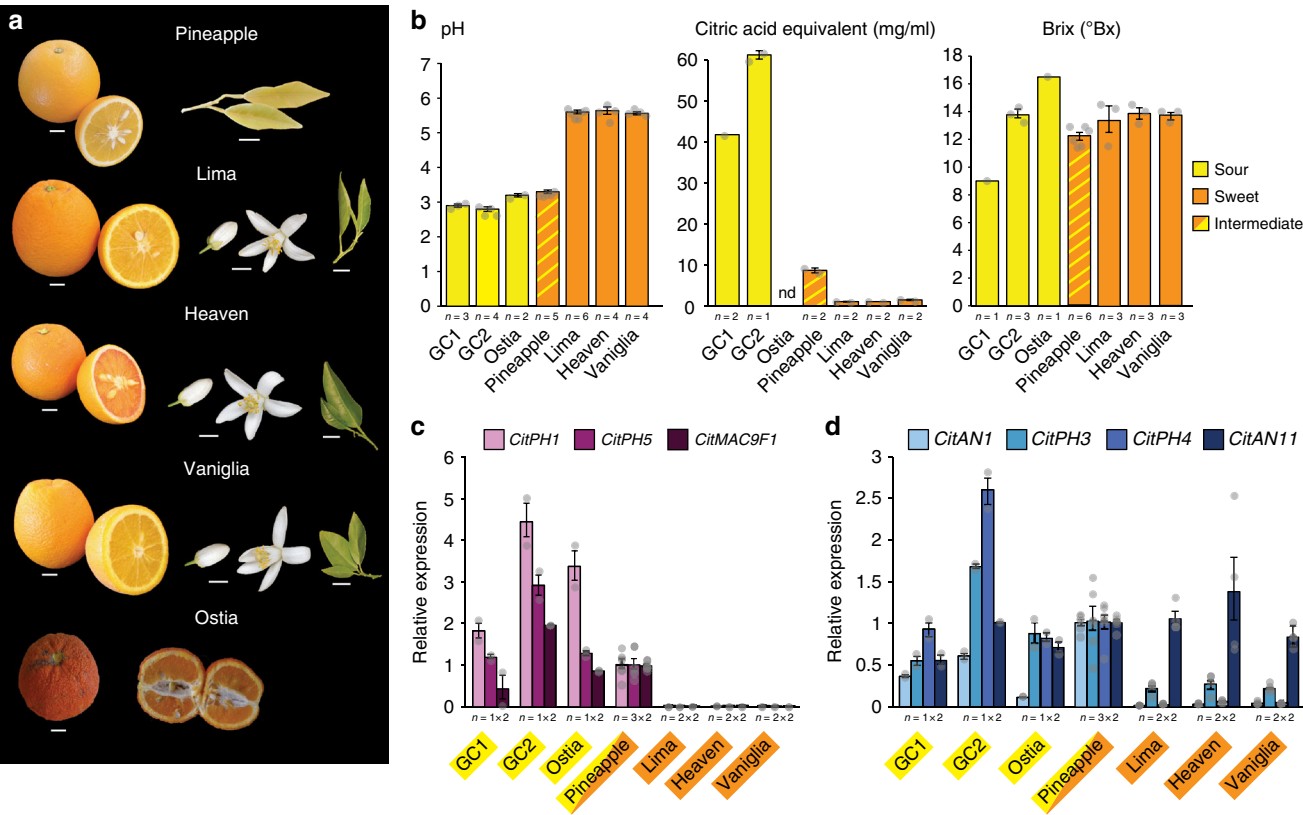

**Fig. 5** Analysis of sweet and sour orange varieties. **a** Fruits, flower, and leaf phenotypes of the analyzed varieties. Size bars represent 1 cm. **b** pH, titratable acid, and soluble solid content (°Brix) in juice from full colored mature fruits (mean ± SE; $n \geq 3$). Note that titratable acid was not determined for Ostia (nd). **c** Real-time RT-PCR of *CitPH1*, *CitPH5*, and *CitMAC9F1* mRNA in juice vesicles from full colored mature fruits. **d** Real-time RT-PCR of *CitAN1*, *CitPH3*, *CitPH4*, and *CitAN11* mRNA in juice vesicles from full colored mature fruits. Values in **c**, **d** are mean ± SE; $n$ = number samples from different fruits × number of technical replicates of each. Source data are provided as a Source Data file

does not contribute to the taste differences (Fig. 5b). The high pH of 'Lima', 'Vaniglia', and 'Orange of Heaven' fruits correlates with strongly reduced amounts of *CitPH1*, *CitPH5*, and *CitMAC9F1* mRNAs and approximately four-fold less *CitPH3* mRNA as compared to the low-pH fruits from 'Pineapple' orange and the trees from Ostia and Gran Canaria (GC1, GC2), possibly due to the strongly reduced levels of *CitAN1* and *CitPH4* mRNA (Fig. 5c, d). RNA-seq data revealed that *CitPH5*, *CitPH4*, and *CitAN1* mRNAs are also downregulated in the 'Sucarri' and 'Bintang' sweet oranges[35].

The low-pH oranges from Gran Canaria (GC1 and GC2) and Ostia all have two *CitAN1* alleles, which differ for a few SNPs and the presence/absence of the same transposon insertion as in *CitAN1Fso* and *CitAN1Aso* (Supplementary Figs. 6–7; Fig. 4). The *citan1* allele(s) of the high-pH 'Lima' and 'Orange of Heaven' oranges are disrupted by the insertion of a 6.9-kb transposable element containing 39-bp inverted repeats (MudR-like element) in intron 2. The *citan1* allele(s) of 'Vaniglia' are disrupted by the insertion of a 5.3-kb copia-like TCS1 retrotransposon in exon 6 (Fig. 4). The downregulation of *CitPH4* is most likely due to an independent mutation, as *CitPH4* expression in lemons is independent from *CitAN1* (Fig. 1).

No relevant differences in the expression of *CitAN11* were detected in fruits of low- and high-pH orange varieties, while *CitPH3* is low expressed in high-pH oranges as compared to the low-pH ones, consistent with its (partial) regulation by CitAN1 and CitPH4[27]. Because the *CitPH4* alleles from these varieties contain no mutations in the coding sequence (Supplementary Fig. 10), it is conceivable that transcription of *CitPH4* (and

possibly *CitAN1*) transcription is reduced by additional mutations in higher-rank regulator(s).

These data show that CitPH1 and CitPH5 are also in oranges responsible for hyperacidification in juice vesicles and that independent mutations in *CitAN1* and an (unknown) upstream regulator of *CitPH4* have been selected to obtain fruit varieties with decreased acidity.

**Analyses of acidless pummelos and rangpur limes.** We further broadened our survey to a group of pummelo accessions (Fig. 6a). Pummelos (*C. maxima*) are non-hybrid citrus fruits[1,3]. They are similar to large grapefruit, variable in sweetness, and native of Southeast Asia but now grown in tropical and subtropical areas all over Asia and the Pacific Islands as well as in California and Florida.

'Pin Shan Kong Yau' (PSKY) and an unnamed 'Kao Panne' pummelo (Punk) are acidic pummelos with blond fruit flesh (Fig. 6a), whereas the unrelated Siamese pummelo P2240 (also known as 'Siamese sweet') is an acidless pummelo with blond flesh that is commonly used in breeding to reduce acidity in the progeny. The "sweet" (non-sour) taste of P2240 fruits correlated with reduced acidity and titratable acid content of the fruit juice and reduced *CitPH1*, *CitPH5*, and *CitMAC9F1* mRNA levels in juice vesicles, as compared to the low-pH PSKY and Punk fruits (Fig. 6b, c). In the pink and mildly sweet-tasting fruits of 'Chandler', a hybrid of *P2240* and an acidic accession (P2241), the juice pH as well as *CitPH1*, *CitPH5*, and *CitMAC9F1* mRNA levels are similar to those of the low-pH PSKY and Punk fruits. This indicates the acidification of pummelo fruits also depends on

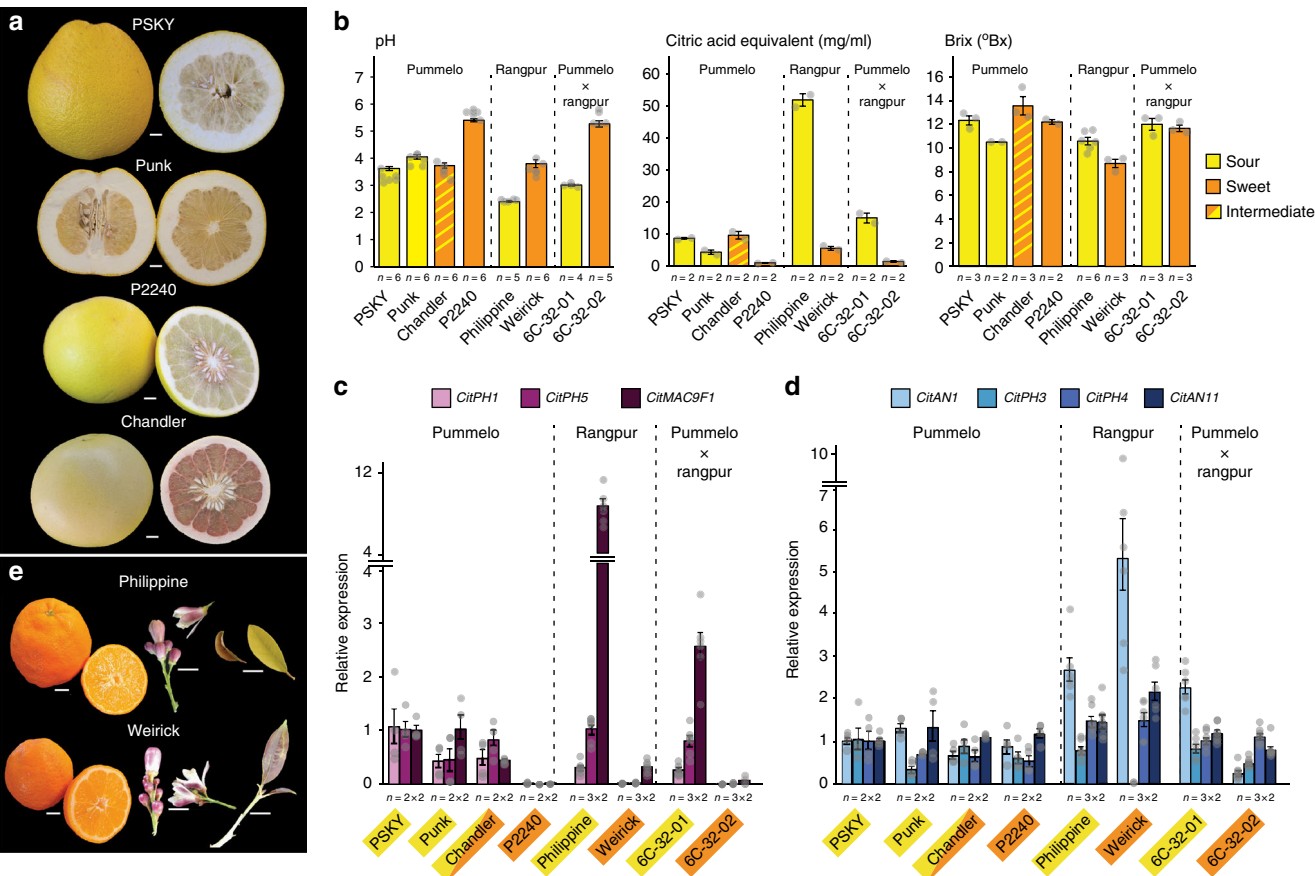

**Fig. 6** Analysis of sweet and sour pummelo and rangpur lime varieties. **a** Fruits of the analyzed pummelo varieties. Size bars represent 1 cm. **b** pH, titratable acid, and soluble solid content (Brix) in juice from the analyzed varieties (mean ± SE). **c** Real-time RT-PCR of *CitPH1*, *CitPH5*, and *CitMAC9F1* mRNA in juice vesicles. **d** Real-time RT-PCR of *CitAN1*, *CitPH3*, *CitPH4*, and *CitAN11* mRNA in juice vesicles. **e** Fruits, flower, and leaf phenotypes of the analyzed rangpur limes. Size bars represent 1 cm. Values in **c**, **d** are mean ± SE; n = number samples from different fruits × number of technical replicates of each. Fruits analyzed in **b**–**d** were mature with a slight green blush (pummelos) or full colored mature (rangpurs and rangpur × pummelo hybrids). Images of Punk and 'Chandler' fruits were provided by and reproduced with permission of the UC-Riverside Citrus Variety collection. Source data are provided as a Source Data file

*CitPH1* and *CitPH5* and that the causative mutation(s) that reduce the acidity and expression of *CitPH1* and *CitPH5* in P2240 fruits (are) recessive.

Since the reduced acidity of P2240 fruits is associated with downregulation of at least three genes (Fig. 6c), the causative mutation(s) most likely affect one of more upstream transcription activators. Because the *CitAN1*, *CitPH3*, and *CitPH4* alleles of *P2240* express in juice vesicles similar mRNA amounts as those from Punk and 'Chandler' (Fig. 6d), and because their coding sequences do not contain polymorphisms with obvious negative effects on protein activity (Supplementary Figs. 11 and 15), the causative mutation likely affects an unknown transcription factor that operates in concert with CitAN1, CitPH3, and CitPH4, which may be encoded or controlled by an unknown locus (*acitric*) within 1.2 cM of the RFLP marker RFZ20 (ref. [41]), which maps to genome sequences on chromosome 2.

The rangpur limes (*Citrus limonia*) 'Philippine' and 'Weirick' have flowers and leaves with anthocyanin pigments (Fig. 6e). While soluble solid content (Brix) is similar, the acidity and amount of titratable acid is reduced in juice from sweet tasting 'Weirick' fruit, compared to the sour fruits from 'Philippine' (Fig. 6b). This correlates with strongly reduced mRNA expression levels of *CitPH1*, *CitPH5*, and *CitMAC9F1*, most likely due to reduced activity of an upstream transcription factor (Fig. 6c). *CitAN1* and *CitPH4* transcripts accumulate at similar levels in

sour 'Philippine' and sweet 'Weirick' juice vesicles, whereas *CitPH3* mRNA was essentially abolished in 'Weirick' fruits (Fig. 6d). PH3 is in petunia, like other WMBW complex components (AN1, PH4, AN11), essential for *PH1*, *PH5*, and *MAC9F1* expression[27], suggesting that the strongly reduced *CitPH1* and *CitPH5* expression in 'Weirick' fruits is most likely caused by abolished *CitPH3* expression. We found no obvious defects in the coding part of the 'Weirick' *CitPH3* allele (Supplementary Fig. 16) or any large rearrangements in its promoter that may account for the reduced *CitPH3* transcription or mRNA processing. Hence, we infer that the reduced acidity and titratable acid is most likely due to a mutation in an (unknown) upstream regulator(s), which strongly reduces the expression of *CitPH3* and its target genes *CitPH1*, *CitPH5*, and *CitMAC9F1*.

The hybrid 'Weirick' × P2240 (7D-76-03) had sour fruits with high titratable acid, indicating that loss of acidity in 'Weirick' and P2240 is due to recessive mutations in distinct genes. Since 7D-76-03 is no longer available, we analyzed progeny with sour (6C-32-01) and sweet fruits (6C-32-02) from the cross 'Chandler' × 7D-76-03 in which segregation of acidity is observed. Whereas soluble solid content (Brix) is similar in both fruits, juice acidity, titratable acid content, and the expression of *CitPH1*, *CitPH5*, and *CitMAC9F1* are all reduced in 6C-32-02 fruits compared to 6C-32-01. The latter is probably due to mutations originating from

P2240 rather than from 'Weirick', as *CitPH3* expression in 6C-32-02 is strongly increased compared to 'Weirick' (Fig. 6d).

## Discussion

How juice cell vacuoles in sour citrus fruits can be acidified to such an extreme extent is a long-standing question, as vacuolar proton pumps capable of generating the required steep pH gradient across the tonoplast were unknown. We have shown that juice vesicles of *Citrus* varieties with acidic (low pH) fruits express *CitPH1* and *CitPH5*, encoding two interacting P-ATPases that constitute a vacuolar proton pump, while *CitPH1* and *CitPH5* expression levels are drastically decreased in fruit varieties with reduced acidity (high pH). The downregulation of *CitPH1* and *CitPH5* in distinct fruits results from independent mutations in multiple genes required for *CitPH1* and *CitPH5* expression, such as, for example, *CitAN1*, or upstream regulators thereof, suggesting that these are the causative mutations for the loss of acidity rather than being linked to them. These results indicate that the long-sought vacuolar proton pump is a P-ATPase complex, encoded by *CitPH1* and *CitPH5*, previously identified for its role in the pigmentation of flowers and seeds. The coupling ratio of P-ATPases ($H^+$/ATP = 1), the sensitivity of PH1/PH5 activity to vanadate, and its insensitivity to bafilomycin[28] fit perfectly with the biochemical properties of the proton-pump activity that is expressed in sour (low pH) fruits but absent in sweet fruit varieties[8,12–15].

While our results show that the large pH differences between acidic (low pH) and acidless (high pH) varieties within a *Citrus* group (lemons, oranges, or pummelos) are due to differences in *CitPH1* and *CitPH5* expression, the cause(s) of the much smaller pH variations between acidic (non-mutant) varieties of different *Citrus* groups or the even smaller pH differences within a group remain unclear. Given that the analyzed varieties are not isogenic and not grown under identical circumstances, such small pH differences may originate from small differences in the expression of CitPH1/CitPH5 or other transporters, like CitSO or various vacuolar antiporters that import solutes in exchange for protons[17].

It is noteworthy that fruits with strongly reduced *CitPH1* and *CitPH5* expression all contain reduced amounts of titratable acids, which is in *Citrus* mostly citric acid. This provides in vivo support of biochemical data, which indicated that most of the citrate transport into vacuoles is driven by the $H^+$ gradient across the tonoplast ($\Delta pH$), while only a small part relies on ATP-driven transporters[9]. Hence, we infer that CitPH1 and CitPH5 promote sour taste by (i) hyperacidifying vacuoles resulting in low pH of the fruit (juice) and by (ii) generating the steep pH gradient required for the import and sequestration of citrate and conjugate bases of other acids into the vacuole. The latter increases pH-buffering capacity of the juice and prevents the juice pH from being neutralized by saliva before the low pH can be sensed.

*Citrus* varieties have been subject to cultivation and selection for several thousands of years[3]. Our data show that citrus varieties with reduced acidity arose multiple times independently in different citrus lineages through mutations disrupting the expression of genes that encode transcription activators of *CitPH1* and *CitPH5*. The finding that the *citan1Fsw* and *citan1Msw* alleles contain several SNPs suggests that the inactivating 3′ deletion and, hence, varieties with reduced acidity, arose in the distant past, possibly hundreds if not thousands of years ago during early stages of *Citrus* domestication. It is, however, difficult to give a precise timing without a better estimate of mutation rates in (inactivated) *Citrus* genes that are not under selection.

Inspection of public RNA-seq data[42] indicates that *Malus domestica* homologs of *PH1* (MDP0000319016) and *PH5* (MDP0000303799) are expressed in developing apples and RT-PCR data[30] indicate that the *Vitis vinifera* homologs *VvPH1* and *VvPH5* are expressed in developing grape berries. Hence, *PH1* and *PH5* and genes encoding upstream transcription activators are likely to be important determinants of the acidity and taste in many other fruits besides *Citrus*.

Taken together, our genetic data show that a vacuolar proton pump consisting of the P-ATPases PH1 and PH5 is required for the hyperacidification of vacuoles in juice vesicles and the very sour taste of Citrus fruits and juices and that over thousands of years of *Citrus* breeding "sweet" (non sour) tasting varieties were obtained many times via independent mutations in distinct transcription regulators driving *CitPH1* and *CitPH5* expression. This opens the way to develop molecular markers for fruit acidity and taste to speed up the breeding in *Citrus* and other fruit crops, most of which are trees or shrubs with long generation times.

## Methods

**Plant material**. Lemon, sweet orange, pummelo, and rangpur lime fruits were collected from trees belonging to the Citrus Variety Collection or from other citrus orchards on the Agricultural Experiment Station at the University of California, Riverside, CA. Sour oranges were collected from a tree in Ostia, Italy (Ostia orange) and from trees grown in the botanical garden Jardin Botanico Canario Viera y Clavijo (GC1) and from the town Agüimes (GC2) in Gran Canaria, Spain. For transient expression assays and isolation of protoplasts, we used petals from the petunia line V74 (*ph4* mutant) grown under normal greenhouse conditions.

**Fruit taste parameters**. Vesicles were excised from the fruits and frozen in liquid nitrogen. After grinding 800 mg of frozen pulp and dissolving in 6 ml distilled water, pH was immediately measured with a pH meter (Consort P901). Brix was determined by directly reading the juice on a refractometer (Marius, Poland). The acid content (expressed as citric acid equivalents) was evaluated by titration using sodium hydroxide (0.1 M) and a phenolphthalein pH indicator. To deliver and measure the volumes, we use a volumetric buret.

**Citrus homologs of petunia AN and PH genes and melon SO**. To identify *Citrus* homologs of petunia and melon genes, we searched *Citrus* genome sequences at Phytozome (https://phytozome.jgi.doe.gov/pz/portal.html) and *Citrus sinensis* Annotation Project (http://citrus.hzau.edu.cn/orange/index.php) with BLAST and confirmed homology by phylogenetic analysis[30]. Protein sequences were aligned using MUSCLE, and after curation by GBLOCKS phylogenetic trees were constructed with maximum likelihood (PHYML) using online tools[43]. For comparison of DNA and proteins, sequences were aligned with Clustal-Omega (https://www.ebi.ac.uk) and MUSCLE (http://phylogeny.lirmm.fr/phylo_cgi/index.cgi) respectively, optimized by hand using Aliview[44], and colored using BOXSHADE (https://embnet.vital-it.ch/software/BOX_form.htm).

*CitAN1*, *CitPH4*, *CitPH3*, and *CitAN11* alleles from citrus varieties (Supplementary Table 2) were isolated by PCR with primers amplifying the entire genomic sequence (for *CitAN1* gDNA primers 7642 and 7590, for *CitAN1* cDNA primers 7642 and 6592, for *CitPH4* gDNA and cDNA primers 6572 and 6573, for *CitPH3* cDNA primers 6611 and 6612, for *CitAN11* cDNA primers 6613 and 6615). PCR fragments were then directly sequenced or cloned into pDONR P1-P2 by Gateway cloning system. *CitSO* cDNAs were amplified by RT-PCR from juice vesicle RNA using primers 8932 and 8933.

To isolate *CitAN1* promoters, PCR fragments were generated using primers covering the 4-kb promoter (primers 7674 and 7473) and used directly for sequencing.

To isolate the flanking sequences of the *CitAN1* gene, we designed primers to amplify fragments (1000 bp) of the *FAR-like* gene located 29 kb upstream of the start of the *AN1* coding sequence (primers 8066 and 8068) and of the *TFIIH* gene (570 bp) located 617 bp downstream of the *AN1* stop codon (primers 8063 and 8065). PCR fragments were directly sequenced and nucleotide polymorphisms were depicted using the IUPAC code. Primer sequences are shown in Supplementary Tables 3–7.

**DNA and RNA isolation from Citrus vesicles**. For DNA and RNA extraction, 800 mg of frozen vesicles were ground in liquid nitrogen and 7.5 ml of preheated (65 °C) extraction buffer (2% (w/v) CTAB, 2% (w/v) PVP (molecular weight 30,000–40,000), 25 mM EDTA pH 8.0, 2 M NaCl, 100 mM Tris-HCl (pH 8.0), 2% β-mercaptoethanol) was added to the frozen powder. To (re)adjust the solution to pH 7–8, we added for citrus vesicles from very acidic fruits (pH 2–2.5) approximately 800 μl 1 M Tris/HCl pH 9.0 and for less acidic or acidless fruits 400 μl Tris/HCl pH 9.0 and verified the pH using pH paper. The samples were then incubated at 65 °C for 15 min and extracted twice with chloroform–isoamyl alcohol (24:1), precipitated with 2-propanol, resuspended in sterile water, extracted with phenol/

chloroform (1:1), precipitated with NaOAc and 2-propanol, washed in 70% ethanol, and resuspended in RNAse free water. Total RNA was precipitated using 1 volume of 4 M LiCl. The RNA pellet was washed with 70% ethanol, dissolved in sterile water, and quantified by measuring optical density (OD)$_{260/280}$. To ensure the absence of genomic DNA, we performed a DNAse treatment.

Total DNA was obtained from the supernatant left after LiCl precipitation of RNA by precipitation with 2-propanol, washed with 70% ethanol, dissolved in water, and checked by agarose electrophoresis and OD$_{260/280}$.

**Expression analysis**. To identify the truncated *AN1* transcript in *Fsw* juice vesicles, we amplified the 3′ cDNA ends by 3′-RACE (5′/3′-RACe KIT 2nd generation; Roche). RT products from *Fso* and *Fsw* *AN1* were amplified with a primer complementary to the 5′ untranslated region (UTR) (primer 6577) and an adaptor primer complementary to the poly(A) tail (primer 64a). Two nested PCRs were then performed using gene-specific primers: nested1 PCR: primer 6577 designed on the 5′ UTR and adaptor primer 65a (tail adaptor); nested2 PCR: primer 6579 designed on the exon 5 sequence and adaptor primer 65a.

RT-PCR analysis of *CitPH1*, *CitPH5*, *CitAN1*, *CitPH3*, *CitPH4*, *CitAN11*, and *CitACTIN* were performed as described previously[24,26,45], using primers shown in Supplementary Table 3. cDNA products were amplified using primers specific for *PH1* (primers 6524 and 4383), *PH5* (primers 6530 and 6529), *AN1* (primers 6623 and 6593, for the *Fsw* allele: primers 6623 and 6592), *PH3* (primers 6538 and 6540), *PH4* (primers 6544 and 6546), *AN11* (primers 6614 and 6615), and *ACTIN* (primers 58a and 59a). All primer sequences are reported in Supplementary Table 3.

Quantitative RT-PCR was performed with an QuantStudio 3 Real-Time PCR System (Thermo Fisher Scientific) using the SensiMix (Bioline QT650–05) as described before[27], using primers shown in Supplementary Table 8. Relative expression was calculated by normalizing against *CitANKYRIN*, *CitANNEXIN2*, and *CitRIBOSOMAL PROTEIN S10*, which are the most constantly expressed genes in broad range of tissues and *Citrus* species[33], and *CitACTIN11*. For 'Schaub', 'Amber' sour, 'Amber' sweet, Sweet lemon, GC1, GC2, and Ostia, one biological replicate was analyzed and for all other varieties two or three fruits. For each fruit, two quantitative PCR reactions (technical replicates) were performed.

**Gene constructs and transient expression in protoplasts**. To generate *35S:CitAN1Fso cDNA*, the 2.2-kb full-size cDNA *AN1* was amplified from *Fso* cDNA with primers 6623 and 6593 and used to generate a Gateway Entry clone by BP reaction with pDONR P1-P2 (Gateway system; Life Technologies, Invitrogen, Carlsbad, CA, USA) and then recombined into the pK2GW7.0 overexpression vector.

*35S:CitAN1Fsw cDNA* construct was obtained by amplifying the PCR fragment of *Fsw* cDNA, with primers 6623 and 6595 and cloning the 1.3-kb cDNA fragment in pDONR P1-P2. The insert was then recombined into pK2GW7.0 by Gateway recombination. The *35S:GFP-CitPH4Fso* construct was generated by amplification of the *PH4Fso* cDNA with a 3′-RACE PCR primer (6572) and a 65a (oligo dT), cloning in pDONR P1-P2, and recombining in pK7WGF2.0 by Gateway system. The *35S:GFP-CitPH4Fsw* construct was obtained similarly from cDNA of *Fsw* juice vesicles.

Promoter:RFP constructs of the target genes were generated as follows: a 3.3-kb *PH1* promoter fragment was amplified with primers 7326 and 7328, cloned in pDONR P1-P2, and recombined into pWSK by Gateway system; a 3-kb *PH5* promoter fragment was amplified with primers 7194 and 7195 and cloned with the same procedure. In *35S:AtAHA10-GFP*, the 35S promoter drives expression of a translational fusion of the coding sequence of the AHA10 gene (including all introns) and GFP[30,32]. All primer sequences are reported in Supplementary Table 9. For isolation and transformation of petunia protoplasts[28,31,32], chopped corollas of 15 open flowers (stage 6–7) from line V74 (*ph4*) were incubated in the dark for 16 h at room temperature in 0.2% (w/v) macerozyme R-10, 0.4% (w/v) cellulose R-10 in TEX buffer (3.1 g/l Gamborg's B5 salts (Sigma-Aldrich), 500 mg/l 4-morpholineethanesulfonic acid (MES), 750 mg/l CaCl$_2$, 250 mg/l NH$_4$NO$_3$, 136.9 g/l sucrose, pH 5.7) and centrifuged for 10 min at 700 RPM. The floating protoplasts were recovered by removing the underlying layers, washed twice in TEX buffer, and left for 2 h at 4 °C. After centrifugation (10 min, 700 RPM), floating protoplasts were recovered and resuspended in 5 ml MMM solution (0.1% (w/v) MES, 0.5 M Mannitol, 15 mM MgCl$_2$). For transformation, 300 μl protoplast suspension, 20 μg plasmid DNA, and 300 μl PEG solution (0.4 M Mannitol, 0.1 M Ca(NO$_3$)$_2$, 40% (w/v) Polyethyleneglycol 4000, pH 8) were mixed, and after 1 min 2 ml of TEX buffer was added. After 2 h at room temperature, 5 ml W5 buffer (154 mM NaCl, 125 mM CaCl$_2$, 5 mM KCl, 5 mM glucose) was added and protoplasts were pelleted for 5 min at 700 RPM, resuspended in 2 ml TEX buffer, and kept for 16 h in the dark at room temperature and then analyzed by confocal microscopy and RNA isolation.

Total RNA was extracted from protoplasts using the NucleoSpin® RNA II Kit (Macherey-Nagel, Düren, Germany). In the expression analysis after transient transformation, *AHA10* mRNA was used as reference to normalize the data.

GFP, RFP, and anthocyanin fluorescence were imaged with a LSM Pascal Zeiss confocal microscope[28,31,32].

**Reporting Summary**. Further information on experimental design is available in the Nature Research Reporting Summary linked to this article.

## Data availability
Identifiers for genes in distinct *Citrus* species and varieties can be found in Supplementary Table 2. Sequence data generated in this study have been deposited in NCBI-Genbank and are accessible under accession numbers MH843936–MH843962 (cDNAs of *CitSO* and *CitPH3*), MH885854–MH885946 (genomic DNAs of *CitAN1*, *CitPH4*, *CitPH3*, *CitFAR*-like, *CitTFIIH-like*), and MH898434–MH898465 (cDNAs of *CitAN1*, *CitPH4*, *CitAN11*, and *CitPH3*). A reporting summary for this article is available as a Supplementary Information file. Source data underlying Figs. 1–3, 5, and 6 and Supplementary Figs. 1, 2, 5 and 13 are provided as a Source Data file. Other data and biological materials are available from the authors on request. Requests for fruits from the Riverside Citrus Variety Collection should be addressed directly to Mikeal Roose [roose@ucr.edu]. Availability is dependent on flowering and fruiting season and on the recipient providing an appropriate import permit if required.

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

## Acknowledgements

Thanks to Toni Siebert and the UC Riverside Citrus Variety Collection for photos and for fruit and leaves and colleagues at the van Leeuwenhoek Center of Advanced Microscopy at University of Amsterdam for the use of microscopy facilities. This work was supported by fellowships to P.S. (iMove, 13001124001502512520) and S.L. (China Scholarship Council 201406910001). C.T.F. and M.L.R. were supported by the USDA National Institute of Food and Agriculture (Hatch project CA-R-BPS-7660-H). Any opinions, findings, conclusions, or recommendations expressed in this publication are those of the author(s) and do not necessarily reflect the view of the National Institute of Food and Agriculture (NIFA) or the United States Department of Agriculture (USDA).

## Author contributions

P.S., S.L., C.E.S., and M.B. performed experiments. C.T.F. and M.L.R. suggested varieties for examination, collected plant material, and provided photos and background knowledge on *Citrus* varieties and genetics. P.S., C.E.S., M.B., F.M.Q., and R.K. analyzed data. P.S., F.M.Q., and R.K. wrote the manuscript. All authors commented on the manuscript.

## Additional information

**Competing interests:** The authors declare no competing interests.

