## [Peer Review File · Nature Communications]

Reviewers' comments:

Reviewer #1 (Remarks to the Author):

The manuscript by Strazzer and collaborators entitled "Flower color genes make citrus fruits sour" presents a very interesting study that shows that a group of genes known to regulate the color in petunia flowers through the expression of a P type H⁺-ATPase complex (encoded by PH1 and PH5) at the tonoplast of petal cells is also involved in the mysterious vacuolar acidification capacity in citrus fruits. The intrinsic difficulty of testing at the molecular level the activity of the candidate citrus homolog genes identified by in silico search was elegantly overcome by the authors. The experiments to monitor the activation of CitPH1 and CitPH5 promoters by CitAN1 and CitPH4 using transient expression of different chimeras in petal epidermis of petunia ph4 mutated flowers is sound (Fig. 2). The authors made a thorough characterization of the expression of the above genes in several sweet and sour varieties of lemons, oranges and pummelos. Overall the manuscript is very well organized and the experiments provide answers that support the authors hypothesis. Furthermore, the presence of PH1 and PH5 homologs in developing apples and grape berries suggest that this group of genes could be involved in acidity regulation of many more fruits in the plant world. The potential biotechnological applications of these findings using mutagenesis via CRIPR are promising.

Reviewer #2 (Remarks to the Author):

Review by Manuel Talon.

I agree with the authors that the elucidation of the mechanisms regulating citrus fruit acidity is a very interesting long-standing question that matters not only to citrus researchers. Fruit acidity is a biochemistry challenge of relevance to practically all fruit crops and to other fields and areas even with implications for the food industry and for the public in general.

I also agree the major conclusion of this work i.e., that the Cit PH1 and Cit PH5 ATPases are major factors determining citrus fruit acidity and that this is a key advance in the knowledge of acid accumulation in fruits. Plant material and Methods are in general solid (but see below).

This manuscript, therefore, reports on interesting original findings that deserve publication.

I need, however, some explanations in several passages of the text. But, first of all I miss final messages summarizing main conclusions, especially after a hard (non-friendly) description of the results. The reader is frequently confused with the names and the associated phenotypes of the several citrus varieties. I guess there is not too much we can do about it, except may be to add explicative sentences in the places where these have not been provided, summarizing the main points driving us to the final message. I also miss this final message, a final conclusion in the discussion.

1) Acidity vs acid taste. Lines 300-307. I find hard to understand from the explanations provided in text the difference between both concepts. Furthermore, this is disturbing when the "intermediate" phenotypes are first introduced (Amber sweet, 175-178, Pineapple, 210-213, Chandler, Fig. 6b) since the reader may think that these varieties do not fit the main hypothesis. However, all these 3 varieties have sweet taste and similar acidic behavior, expression of the ATPases and of CitAN1, and therefore, it is reasonable to suggest that the acid taste is also determined by the acid's profile, i.e. the amount and nature of different weak acids (increased TA), presupposing the occurrence of additional factors other than CitPH1 and CitPH5 controlling the acid profile. A more specific explanation should be given in here.

2) I find reasonable the thinking that Faris and Amber varieties are graft chimeras. But I do not

find an explanation of how we get CitAN1 homozygosity in Faris sweet from a Millsweet cell with two different CitAN1 alleles that migrates from L2 to L1 (or viceversa). It would be interesting to add comments in this sense.

3) It also disturbs me the use of sap for the determinations on Faris (Fig. 1) while in all other varieties juice is used. There is no mention in material and methods and furthermore phloem sap is always alkaline.

4) Methods. Since several of the measured physiological parameters are a function of time I guess we need to add a comment on the period of sampling of the different fruits. Furthermore, it is not usual to make comparisons with only 3, 2 or even 1 single fruit. Please, revised carefully the data because there are error bars in columns with $n = 1$ and please add a comment explaining or justifying why this small number of replicates is enough in this kind of measurements.

5) It appears that there are repeated data in Fig. 1 c and e, and in Fig. 3 c and d, that show the same results about relative expression of several genes in Frost Lisbon. Please, fix this.

Other changes:

Cells of juice vesicles or pulp cells... instead of juice cells?

Line 38. Occurs?

Lines 82-83. CsSO or CitSO

Line 97. Wrong reference.

Lines 116-117. Fsw?

Line 159. Fig. 1e?

Line 181. Sour Amber?

186. Fig. 3d? 'Amber'

204. Bold letters?

256. Fig. 6c?

263. Wrong cite format

Supplementary Figure 4. Reference number 3 does not report on RNA-seq data.

Fig. 1. It would be convenient to specify which structure in panel f corresponds to Fso and Fsw. Also, if panel b is dealing with pulp cells and not with sap, then the taste colour code should be added as in the rest of figures.

Fig. 5 and 6. Figures 1 and 3 show a nice mnemotechnical order in the panels that helps the reader; from left to right (sour, intermediate, sweet) but Figures 5 and 6 follow another order, any reason?

Reviewer #3 (Remarks to the Author):

This manuscript is well-prepared manuscript and provides evidence for the role of P-type ATPase genes and their regulators involved in the accumulation of organic acid in Citrus fruit. This study is interesting and will be helpful for genetic improvement of fruit acidity in Citrus. Overall, the finding of flower color genes make citrus fruit sour was well supported in 'Faris' sweet lemons and other lemons cultivars, but there are some shortcomings when this finding was extended to oranges, Acidless pummelos and rangpur limes. The detail of the concerns are listed as follows:

--Figure 3, 'Frost Lisbon' was used for real-time PCR analysis. It would be better if acidity and sweetness of this cultivar were added in figure 3b.

--Figure 5, i) this manuscript clearly indicated that flower color genes make citrus fruits sour, and figures 1 and 3 supported this finding very well. However, the flower and its color of Pineapple and

Ostia were missing in figure 5a, and their flowers were expected to be colored. ii) the inconsistency between pH value and titratable acid content, especially for Ostia. Is there any significant difference between the pH of Ostia and those of three other cultivars low pH, Pineapple, GC1, and GC2. Iii) gene expression profiles showed that CitPH4 may be more important than CitAN1.

--Figure 6, i) Philippine and Weirick were quite different in pH and titratable acid content, but their flower color are very similar. Moreover, for these two accessions, the expression of PH1 and PH5 was not well correlated with organic acid content as well? Why? Ii) AN1 showed the highest level of expression in Weirick, but PH1 and PH5 showed an extremely low expression. It is in contrast with the finding that AN1 was an activator of PH1 and PH5.

--Discussion, the second paragraph, citrate accumulation was mentioned to be associated with fruit sour in citrus. However, other organic acids such as malate are also present in citrus fruit although citrus fruits predominately accumulate citrate, and some cultivars may accumulate higher level of malate, instead of citrate. In addition, it is unclear whether the organic acid components were partially related to the inconsistency between pH and titratable acid content.

Reviewer #4 (Remarks to the Author):

This paper relates to the contribution of two proton transporter in the acidification of fruit of the Citrus family, and to the identification of the CsAN1 transcription factor as the mutated locus controlling the mutant non-sour citrus genotypes studied. The research was carried out by one of the leading research groups studying proton transporters in plants, and they have been joined in this report by one of the leading citrus geneticists with a novel collection of citrus germplasm. The research has taken a strictly genetic approach, combining a unique collection of Citrus accessions and novel mutants representing the range of citrus fruit vesicle acidification, with extensive sequencing and expression analyses.

The most significant and strongest conclusion of the research that that the loss of expression of CitPH1 and CitPH5 in the sweet mutants is correlated with the reduction in acidity, and that the mutation is in the CitAN1 transcription factor which regulates their expression. The authors show this most conclusively in their coexpression reporter gene studies in petunia protoplasts.

Regretfully, research on fruit trees does not allow for the decisive in planta functional expression and complementation studies carried out on annual fruiting plants. Considering the limitations of the perennial tree system with its juvenility requirements one could not expect more.

This is an extremely important contribution to understanding the genetic physiology behind these sweet mutants and points to the contributory role of these transporters in citrus fruit acidity. The special novelty of this, as the authors discuss, is that citrus fruit and petunia flowers share a tissue specific module of TF: functional gene interactions responsible for acidification, AN1:PH1,PH5.

The researchers expand their observations to other Citrus species, including additional lemons, oranges and pummelo/grapefruits, with novel ranges of acidity. Here too, the contribution of PH1,PH5 expression is crucial to acidification, and sometimes, but not always, related to AN1 expression levels. With this expansion of genotypes to the survey the authors conclusively show that PH1 and PH5 contribute to fruit acidification, most likely across the Citrus family, and that the regulation of these two transporters is likely under different molecular genetic control within the family. Such a range of genetic control mechanisms of a phenotype is to be expected for such downstream functional gene expression.

As such, I would recommend that this paper be published in Nature Communications as it makes an important contribution to our understanding of a subject of broad interest: plant evolution of consumable fruit leading to taste differences. However, I suggest that the paper should be revised taking into consideration the comments below.

There are some major comments, followed by minor comments.

Major comments:

1- As indicated by my positive comments, the paper shows that PH1 and PH5 expression loss is probably causal to the low acid levels in the sweet mutations and that these transporters are likely

the major contributors to citrus fruit acidity across the family, including in the non-mutant accessions. However, this does not indicate that quantitative differences in expression of these two genes are causal to the broad genetic variability of acidity among the non-mutant accessions. The conclusions regarding the contribution of PH1 and PH5 expression levels to the range in fruit acidity among non-mutant citrus fruit should be limited for the following reasons:

1- Results of "relative expression" from qRT-PCR from different species, especially as normalized against actin, should be viewed with some apprehension. This is particularly so for relative values of half or even 2-fold. Without absolute read values of NGS with rigorous statistics and replications the "relative expression" leaves much to be desired and differences should be interpreted cautiously. This is especially so when comparing different accessions, and distantly related (or unrelated) lines.

2- Even the harvest stage chosen can be a confounding factor as each accession likely has a somewhat different developmental pattern of fruit development (in fact, some of the differences in acidity between lines may in fact be related to different developmental stages at harvest). It is well known that citrus fruit acidity undergoes major developmental fluctuations. The authors do not describe in the methods section the stage of fruit development (this should be done as descriptively as possible- realizing that flowers were not tagged at anthesis) but it is unlikely, or even impossible, that there is any consistency in fruit stages. In an ideal situation each line would be studied under a number of defined developmental stages in order to extrapolate the contribution of expression levels to acid accumulation. But without this the authors should be conservative with their conclusions.

3- A further indication that the expression levels (at only a single developmental stage) of PH1 and PH5 do not explain fruit acidification across the citrus germplasm is the difference in acidification between lemons and the other acidic non-mutant citrus. Lemons undergo a larger "hyperacidification than do sour oranges and from the qRT-PCR results the expression of PH1 and PH5 at a single developmental stage cannot explain the difference. In fact, a simple correlation between acidity (pH?) and expression of PH1 and PH5 across the non-mutant accessions shows that there is little correlation. This indicates that while PH1 and PH5 are indeed major genes with epistatic control over citrus acidification, their role in being causal to the genetic variability among non-mutant Citrus cannot be ascertained from this study and indications from this study are in fact that there may be little (e.g., PH1 and PH5 expression levels among the different sour orange varieties in Fig. 6).

In conclusion, the major comment is that the paper should be somewhat modified taking the above points into consideration. The clear conclusions relating to the role of these transporters and the mutation in AN in the mutant lines should be separated from the broader picture of acidification in the non-mutant Citrus lines. For the latter, the authors may conclude that the two transporters do indeed contribute but that it cannot be concluded that genetic variability for them contributes to the variability in acid levels within the non-mutant family. Also, it may not be cautious to claim that (line 237) PH1, PH5 are "responsible for hyperacidification" and the term perhaps should be preserved for the extreme acidic sour lemons, as was by Muller et al (1996). Additionally, the authors ruled out the possible contribution of the SOUR transporter to acidification in the mutant lines based on expression levels. While for the other genes studied the authors also sequenced the gene to be certain that only expression levels were of relevance, for the SOUR gene they did not sequence the different alleles. However, the SOUR gene may still be involved since it was shown that the evolution of non-sour in melon was due to a change in gene sequence rather than expression. Although it seems most likely that the AN:PH1PH5 mutation is indeed causal in the mutant citrus, nevertheless the SOUR sequence should be reported as well. Furthermore, it still remains a possibility that SOUR expression levels are related to the genetic variability in the non-mutant citrus. Although this may be out of the scope of the present paper, I imagine the authors still have all the RNA and cDNA samples and a qRT-PCR for the SOUR gene should be relatively painless and might offer some interesting insights.

I would also suggest that the authors modify the title of the paper, "Flower color genes make citrus fruit sour". Although catchy, it is really incorrect, and seems inappropriate for a Nature Communications paper. PH1 and PH5 are not "flower color genes" but as the authors describe in their Li et al 2016 Evolution paper, they are ubiquitous tonoplast P-ATPase transporters. That may

sound less sexy but the title should be a scientific description of the major point.

Minor comments:

1- "brix" should be presented as degrees (Brix⁰), not %, and does not represent the soluble sugar content but rather, as an indication of refraction, all soluble solids. Sugars are the major soluble solids that accumulate and change with development and as such Brix can parallel developmental changes in sugar levels. But since it is not actually sugar the correlation between Brix and sugar is not strong enough to extrapolate that a difference in Brix of 1, for example, is related to a parallel difference in sugar.

2- Perhaps my copy was printed poorly but my Fig. 4 for Sweet Lemon shows one allele with the central 4 exons in light grey, indicating a splice variant, and the second allele is all grey, indicating silencing (?). The results section (line 194) describes otherwise.

3- In Fig. 3d, for the expression of AN1 in Sweet Lemon, which 3' primers were used? Did they distinguish between the deletions?

4- In Fig. 1f, mutant- Indicate what the different dotted lines leading to the 3' end represent in the legend.

In conclusion, I commend the authors on a very significant paper but suggest the following revisions

1- Modification of the narrative and discussion, emphasizing that the research results indicate that the non-sour mutations point to the major role of PH1 and PH5 in citrus acidification but do not imply that they are responsible for the broad genetic variability of acidification in the family.

2- Description of the sampling stages and referral to the limitations of a single harvest stage in such a study.

3- Sequences of the SOUR alleles and expression of SOUR in the other citrus genotypes.

4- Modification of the title to be more descriptive.

5- The 4 minor comments.

Response to Reviewers

Below we outlined point by point how we addressed the comments and suggestion of Reviewers 1-4 . The text of the reviewers is shown in black lettering and our reply in blue.

Note that the line numbers cited by the reviewers refer to the original manuscripts (1st submission), whereas the line numbers cite in our reply refer to the text in which changes have been marked with MsWords track changes.

As Word changes the line numbering in documents with tracked changes continuously, we included the highlighted text at the end of this PDF file.

Reviewer #1 (Remarks to the Author):

The manuscript by Strazzer and collaborators entitled “Flower color genes make citrus fruits sour” presents a very interesting study that shows that a group of genes known to regulate the color in petunia flowers through the expression of a P type H⁺-ATPase complex (encoded by PH1 and PH5) at the tonoplast of petal cells is also involved in the mysterious vacuolar acidification capacity in citrus fruits. The intrinsic difficulty of testing at the molecular level the activity of the candidate citrus homolog genes identified by in silico search was elegantly overcome by the authors. The experiments to monitor the activation of CitPH1 and CitPH5 promoters by CitAN1 and CitPH4 using transient expression of different chimeras in petal epidermis of petunia ph4 mutated flowers is sound (Fig. 2). The authors made a thorough characterization of the expression of the above genes in several sweet and sour varieties of lemons, oranges and pummelos. Overall the manuscript is very well organized and the experiments provide answers that support the authors hypothesis. Furthermore, the presence of PH1 and PH5 homologs in developing apples and grape berries suggest that this group of genes could be involved in acidity regulation of many more fruits in the plant world. The potential biotechnological applications of these findings using mutagenesis via CRIPR are promising.

REPLY: Thank you for these positive comments

Reviewer #2 (Remarks to the Author):

Review by Manuel Talon.

I agree with the authors that the elucidation of the mechanisms regulating citrus fruit acidity is a very interesting long-standing question that matters not only to citrus researchers. Fruit acidity is a biochemistry challenge of relevance to practically all fruit crops and to other fields and areas even with implications for the food industry and for the public in general.

I also agree the major conclusion of this work i.e., that the Cit PH1 and Cit PH5 ATPases are major factors determining citrus fruit acidity and that this is a key advance in the knowledge of acid accumulation in fruits. Plant material and Methods are in general solid (but see below).

This manuscript, therefore, reports on interesting original findings that deserve publication.

I need, however, some explanations in several passages of the text. But, first of all I miss final messages summarizing main conclusions, especially after a hard (non-friendly) description of the results. The reader is frequently confused with the names and the associated phenotypes of the several citrus varieties. I guess there is not too much we can do about it, except may be to add explicative sentences in the places where these have not been provided, summarizing the main points driving us to the final message. I also miss this final message, a final conclusion in the discussion.

REPLY:

(a) We added additional explicative sentences on several point (lines 307-312, 362-364, 405-408).

(b) We added an extra paragraph at the very end of the discussion with a final (overall) message (lines 494-500)

1) Acidity vs acid taste. Lines 300-307. I find hard to understand from the explanations provided in text the difference between both concepts. Furthermore, this is disturbing when the “intermediate” phenotypes are first introduced (Amber sweet, 175-178, Pineapple, 210-213, Chandler, Fig. 6b) since the reader may think that these varieties do not fit the main hypothesis. However, all these 3 varieties have sweet taste and similar acidic behavior, expression of the ATPases and of CitAN1, and therefore, it is reasonable to suggest that the acid taste is also determined by the acid’s profile, i.e. the amount and nature of different weak acids (increased TA), presupposing the occurrence of additional factors other than CitPH1 and CitPH5 controlling the acid profile. A more specific explanation should be given in here.

REPLY: We infer from this comment that it is the relation between low pH, total (titratable) acid content and taste that does not become sufficiently clear, and causes problems to understand “intermediate” phenotypes (Amber sweet, Pineapple etc.) fit in the hypothesis.

- We added an extra sentence at the beginning of the Introduction (lines 31-33), to explain (1) that we perceive a sour taste when acid-sensitive cells in taste buds “feel” a high concentration of free H^+ ions and (2) that this happens when juice has a low pH AND a minimal pH buffering capacity, to prevent that juice pH is immediately neutralized by the saliva, before the taste buds can perceive a high H^+ concentration. Furthermore, we added a sentence (lines 37-39) explaining that the vacuolar accumulation of citrate (which represents the bulk of the organic acids in *Citrus*) contributes to sour taste by increasing pH buffering capacity, not by lowering the pH (quite the contrary: as it is most likely the conjugate base citrate³⁻ that is transported, that actually increases vacuolar pH).
- In the discussion we originally summarized the role of the proton-pump by writing that “PH1 and PH5 are necessary for sour taste but not sufficient”. Apparently that was too short and condensed to be understandable. In a newly written paragraph of the discussion (lines 463-471) we now explain in a few

more (simple) words the relation between pH, titratable acid and eventually perception of sour taste and how CitPH1 and CitPH5 fit in that.

- The comment that ‘Amber’ has an “intermediate” phenotype is most likely due to the ambiguity concerning acidity and acid taste. As far as acidity (fruit pH) is concerned, which is the trait/phenotype that is central here, sweet ‘Amber’ fruits are full wild type in spite of their sweet taste. That is, juice pH is as low as in the acidic (low-pH) lemons sour ‘Amber’, sour ‘Faris’ and ‘Frost Lisbon’ and, consistent with the main hypothesis, mRNAs for *CitPH1* and *CitPH5* are expressed at similarly high levels. For that reason we had introduced both sweet and sour ‘Amber’ at the beginning along with the low pH variety ‘Schaub’. However, we agree that things might be easier to follow if Amber is introduced as last of the lemons, and therefore moved the corresponding text to the end of its section (lines 262-306).
- To make the texts on the Amber fruits easier to follow, we now first explain that sweet Amber fruits have low pH and express (in line with the hypothesis) high levels of CitPH1 and CitPH5, similar to sour Amber and Frost Lisbon fruits, before we explain that the sweet taste of Amber sweet is caused by a reduction in TA, most likely due to a defect in another pathway.
- Regarding the introduction of ‘Pineapple’ and ‘Chandler’, we refer to our reply to the comments of this reviewer on Fig 5 and 6.

2) I find reasonable the thinking that Faris and Amber varieties are graft chimeras. But I do not find an explanation of how we get CitAN1 homozygosity in Faris sweet from a Millsweet cell with two different CitAN1 alleles that migrates from L2 to L1 (or viceversa). It would be interesting to add comments in this sense.

REPLY: It seems that this comment stems from two distinct misunderstandings:

- First, we do not know with certainty that Fsw is homozygous, as we cannot exclude the possibility that Fsw is actually hemizygous and that sister chromosome contains a deletion spanning *CitAN1* and flanking genes. In the original text we wrote:
“We found no sequence polymorphisms distinguishing the Fso alleles of *CitAN1* or the flanking genes *CitFAR-like* and *CitTFIIH-like* (Supplementary Fig. 7), indicating that the genomic *CitAN1* region is in Fso either fully homozygous or hemizygous over a larger deletion spanning *CitAN1*, *CitFAR-like* and *CitTFIIH-like*”.
We assume that “.... no sequence polymorphism distinguishing the Fso alleles...” may put readers (and this reviewer) on the wrong leg that there must be two alleles. To avoid that, we now explain the actual findings and their interpretation more precisely (lines 156-160) by writing that (i) we found no polymorphisms in PCR products amplified from Fso fruits, which indicates that (ii) the genomic *CitAN1* region is either homozygous or, as deletions of multiple genes have been found in citrus, hemizygous over a sister chromosome with a large deletion.
- Second, previous SSR analysis showed that L1 layer of Faris originates from a variety that matched with Millsweet for all 12 tested markers. This suggests that the L1 donor maybe either Millsweet or a close relative thereof. As acidity is more strongly reduced in Faris sweet than in Millsweet (see also Fig 1a and 2a) the most likely candidate is thought to be a relative of Millsweet, possibly Mediterranean sweet lime (see Aprile et al 2011 Acta Hort 892:37–42). In the original version we wrote that the L1 layer of Faris derives from “Millsweet limetta or a related genotype”. We now rephrased that as “ an unknown variety related to ‘Millsweet’ (line 88).
- The first plant containing this 3’ deletion allele appeared very long ago (as inferred from the many polymorphisms in the Msw and Fsw alleles) and was almost certainly heterozygous. Homozygotes only appeared later in segregating progenies originating from selfings and/or crosses between heterozygotes.

It is likely that Millsweet and Faris sweet originate from related siblings from such a segregating population, which happened to be respectively heterozygous and homozygous for the 3' deletion allele. In the paragraphs where we discussed the ancient origin of the 3' deletion allele, we added an extra sentence explaining this (line 249-252).

3) It also disturbs me the use of sap for the determinations on Faris (Fig. 1) while in all other varieties juice is used. There is no mention in material and methods and furthermore phloem sap is always alkaline.

REPLY: We used "sap" as a synonym of "juice" (from fruits, not from phloem). As that apparently creates confusion we replaced "sap" with "juice" throughout the text.

4) Methods. Since several of the measured physiological parameters are a function of time I guess we need to add a comment on the period of sampling of the different fruits. Furthermore, it is not usual to make comparisons with only 3, 2 or even 1 single fruit. Please, revise carefully the data because there are error bars in columns with $n = 1$ and please add a comment explaining or justifying why this small number of replicates is enough in this kind of measurements.

REPLY: This comment touches on several issues, which are discussed one by one below:

- We added a morphological description of the developmental stages of the fruits that were used to the legends of Figs 1, 3, 5 and 6.
- As Citrus varieties are not exactly "isogenic" and their exact history and genetic relation are often unknown, comparison of gene expression patterns between a single acidic (low pH) and acidless (high pH) variety is insufficient to demonstrate a causative relation, as those varieties will differ in many other traits too. Therefore it is essential to compare gene expression patterns in multiple sweet and sour varieties and demonstrate that (1) a particular gene set is consistently up or down regulated in many acidic or acidless varieties, and (2) that such differences originate from independent mutations, rather than a single ancient mutation that now "segregates" among modern varieties, to distinguish between mutation(s) that cause a certain trait and mutations (or polymorphisms) that are linked with the trait. Hence, analyses of single fruits from 10 sour and 10 sweet unrelated varieties makes a much stronger point than 10 replicates from a single sweet and a single sour variety. In other words analysis of even a single fruit from an additional variety (e.g. Ostia) makes a much stronger point than another replicate of a variety for which some data are already available. For this reason we also included data on varieties for which only a single fruit was available.
- The error bars for the $n=1$ columns are calculated from technical replicates. Note that we indicated for all RNA data the number of replicates as $A \times B$, in which A and B stand for the number biological and technical replicates respectively

5) It appears that there are repeated data in Fig. 1 c and e, and in Fig. 3 c and d, that show the same results about relative expression of several genes in Frost Lisbon. Please, fix this.

REPLY: All relative expression data that are shown were determined in one big (final) experiment in which we isolated RNA and analyzed expression levels (qRT-PCR) side by side. The many RT-PCRs that were done on smaller sets of varieties in the earlier stages of this project are not shown/included. The relative expression data in Fig 1 and Fig 3 are all on lemons from different varieties and can therefore be directly compared.

We considered to show these data all in a single graph (instead of two), but found that unsatisfactory, as it makes the description of the results even "harder" or more "unfriendly". Deleting the data on Frost Lisbon from Fig 3 makes it impossible for readers to compare the data on the lemon varieties in Fig 1 with those in Fig 3, which we found also undesirable. Thus, we decided to keep the duplicated Frost Lisbon data in Fig 3 and added a sentence in

the legend of Fig 3 that the Frost Lisbon data from Fig 1 are repeated there for comparison.

Other changes:

Cells of juice vesicles or pulp cells... instead of juice cells?

REPLY: We replaced "juice cells" with "juice vesicles" in most places, and in some places with "juice vesicle cells"

Line 38. Occurs?

REPLY: Yes, "couss" should indeed have read "occurs". We corrected this typo.

Lines 82-83. CsSO or CitSO

REPLY: "CsSO" should indeed be "CitSO". We corrected this typo.

Line 97. Wrong reference.

REPLY: Indeed. The correct citation here is Xu et al 2013 (=ref 40 in original ms), not Wu et al 2014 (=ref 1 in original and revised ms). We corrected this copy-paste error. Note that as a consequence Xu et al 2013 is now ref 35 in the revised ms

Lines 116-117. Fsw?

REPLY: Yes, "Fso" should have read "Fsw" in both lines. Note that we have rephrased these lines (see our reply to comment 2 of this reviewer)

Line 159. Fig. 1e?

REPLY: Yes, indeed. We should have referred to Fig 1e, not to 1d. This is corrected

Line 181. Sour Amber?

REPLY: Yes indeed. We added "sour"

186. Fig. 3d? 'Amber'

REPLY: Yes indeed, we change 3a into 3d

204. Bold letters?

REPLY: Yes that line is a section title. We made it bold.

256. Fig. 6c?

REPLY: We agree and changed Fig 6d into Fig 6c

263. Wrong cite format

REPLY: We note that in Nature journals citations following directly after a number (here: RFZ20) are not indicated with a superscript number but as (ref. XX) to discriminate it from a number Y to the power X. Hence we did not change this.

Supplementary Figure 4. Reference number 3 does not report on RNA-seq data.

REPLY: We agree. We replaced the citation to Wu et al (2014) with Xu et al (2013), as in response to the comment above (on line 97).

Fig. 1. It would be convenient to specify which structure in panel f corresponds to F_{so} and F_{sw}. Also, if panel b is dealing with pulp cells and not with sap, then the taste colour code should be added as in the rest of figures.

REPLY: We agree and (i) added a key for the colors in panel b and (ii) added labels for the *CitAN1*^{F_{so}} and *CitAN1*^{F_{sw}} alleles in panel f

Fig. 5 and 6. Figures 1 and 3 show a nice mnemotechnical order in the panels that helps the reader; from left to right (sour, intermediate, sweet) but Figures 5 and 6 follow another order, any reason?

REPLY: Figs 1 and 3 show data on lemon varieties only, whereas Figs 5 and 6 show data on distinct types of fruits. Fig 5 shows data on sweet and sour oranges, which are traditionally seen as different types, although genome sequencing shows the division is so clear cut. In Fig 5 we moved the bars for the sour oranges (GC1, GC2, Ostia to the left) and bars of the sweet oranges (Pineapple, Lima, Heaven, Vaniglia) to the right. In this way they are still clustered by type (sweet or sour oranges) and are also arranged in order of their acidity (sour, intermediate, sweet). We also adjusted the corresponding text and now first discuss the data on the sour oranges before the sweet oranges, following the suggestion by reviewer 2.

Fig 6 shows data on pummelos, rangpurs and pummelo X rangpur hybrids. To help readers see that more easily, we indicated in the Fig 6 which varieties are pummelos, rangpurs or hybrids. In the set of pummelos we moved the data on Chandler to the left of P2240, to ensure for each fruit type the order is from sour to sweet fruits

Reviewer #3 (Remarks to the Author):

This manuscript is well-prepared manuscript and provides evidence for the role of P-type ATPase genes and their regulators involved in the accumulation of organic acid in Citrus fruit. This study is interesting and will be helpful for genetic improvement of fruit acidity in Citrus. Overall, the finding of flower color genes make citrus fruit sour was well supported in 'Faris' sweet lemons and other lemons cultivars, but there are some shortcomings when this finding was extended to oranges, Acidless pummelos and rangpur limes. The detail of the concerns are listed as follows:

--Figure 3, 'Frost Lisbon' was used for real-time PCR analysis. It would be better if acidity and sweetness of this cultivar were added in figure 3b.

REPLY: The data on acidity and sweetness (does that refer to TA?) and gene expression levels are already shown together in Fig 1. We repeated qRT-PCR data on Frost Lisbon (which are shown in Fig 1) in Fig3 to enable comparison of the relative (!) expression levels across all lemon varieties in Fig 3.

To address a comment of reviewer 2 we added the following sentence in the legend of Fig 3: "Relative mRNA expression levels from 'Frost Lisbon' were taken from Fig 1, and repeated here for comparison."

--Figure 5, i) this manuscript clearly indicated that flower color genes make citrus fruits sour, and figures 1 and 3 supported this finding very well. However, the flower and its color of Pineapple and Ostia were missing in figure 5a, and their flowers were expected to be colored. ii) the inconsistency between pH value and titratable acid content, especially for Ostia. Is there any significant difference between the pH of Ostia and those of three other cultivars low pH, Pineapple, GC1, and GC2. iii) gene expression profiles showed that CitPH4 may be more important than CitAN1.

REPLY: The remarks (i) to (iii) are discussed one by one below:

- (i) In the original title "Flower color genes make citrus fruit sour" flower color genes referred to genes that color petunia flowers, not necessarily citrus flowers. Given that the role of AN1 homologs as activators of anthocyanin synthesis is widely conserved, and because CitAN1 is expressed in flowers it is very likely that

CitAN1 is also involved in pigmentation of citrus flowers. However, for CitPH4 and the structural genes CitPH1 and CitPH5 that is less likely, as their expression in citrus flowers is low (see Supplementary Fig. 5).

- (ii) Titratable acid was not determined for Ostia fruit, which was indicated as “nd” in Fig 5b. We assume that the reviewer interpreted “nd” as “not detectable” as the abbreviation was not specified. We now added to the legend of Fig 5 that nd means “not determined”. We cannot tell from the available data whether the pH of Ostia fruits is significantly different from that of the other low pH oranges. Even if the difference would be significant, the difference is quite small. Given that the trees of different varieties are not isogenic and were not growing side by side such a small difference may originate from (differences in their environment and/or genetic background affecting the activity of other proteins (e.g. H⁺ antiporters, like NHX; see also our reply to Reviewer 4). Hence it is difficult, if not impossible, to correlate the small pH differences to variations in CitPH1 and/or CitPH5 expression. To point this out for readers we added additional text in the Discussion (lines 455-462).
- (iii) The data on AN1, PH4 and PH3 in petunia indicate that these proteins act in a complex (WMBW) and that mutations eliminating any of them cause a strong down regulation of PH1 and PH5 expression and similar shifts of the pH of petal extracts. Hence, it is not possible to say that one is more important than the other for expression of PH1 and PH5. The same holds true for the Citrus genes. Sweet oranges all have transposon insertions in CitAN1 that may well explain the strong reductions in CitAN1 mRNA that we observed. Moreover in the Vaniglia also the coding sequence is disrupted. Hence, these varieties have a clear problem in expressing AN1, which is itself sufficient to strongly reduce PH1 PH5 expression. In these lines expression of CitPH4 is reduced to a similar strong extent as CitAN1, which should reduce PH1 and PH5 expression as well (cf. Fig 2). Hence,

--Figure 6, i) Philippine and Weirick were quite different in pH and titratable acid content, but their flower color are very similar. Moreover, for these two accessions, the expression of PH1 and PH5 was not well correlated with organic acid content as well? Why? ii) AN1 showed the highest level of expression in Weirick, but PH1 and PH5 showed an extremely low expression. It is in contrast with the finding that AN1 was an activator of PH1 and PH5.

REPLY: Comment (i) relates to comment on Fig 5, and may have also been triggered by the (unwanted) suggestion from the original title that fruit acidity is determined by genes that color *Citrus* flowers.

We do not understand the remark about organic acid content and CitPH1/PH5 expression in Philippine and Weirick. Expression of CitPH1 and CitPH5 (and CitMAC9F1) is strongly reduced in Weirick, when compared to Philippine, and, consistent with the hypothesis, pH of Weirick is up and, consistent with the idea that the pH gradient across the tonoplast drives citrate uptake, titratable acid is down.

Comment (ii): CitAN1 expression in Weirick appears indeed higher than in Philippine, but since the difference is less than two-fold, we don't want to conclude much from that (see also our reply to comment 3 of reviewer 4). The key point that the reviewer missed is that expression of CitPH3 is down (essentially abolished completely) in Weirick fruits. Given that PH3, which is a WRKY transcription factor that binds the WD40 component (AN11) of the WMBW complex, is in petunia essential for PH1 and PH5 expression (Verweij 2016; ref 27) then down regulation of CitPH1 and CitPH5 is most likely due to the strongly reduced CitPH3 expression. We have added a few sentences in the results (lines 400-408) to point this out more clearly.

--Discussion, the second paragraph, citrate accumulation was mentioned to be associated with fruit sour in citrus. However, other organic acids such as malate are also present in citrus fruit although citrus fruits predominantly accumulate citrate, and some cultivars may accumulate higher level of malate, instead of citrate. In addition, it is unclear whether the organic acid components were partially related to the inconsistency between pH and titratable acid content.

REPLY: "Titratable acid" refers to the amount of NaOH needed to neutralize the pH of the juice and thus measures all acids in the juice, irrespective of whether it is (mostly) citric or malic acid. The reason to measure TA is that a sour taste of fruit juice (or any other beverage) requires low pH and a certain buffering capacity. In other words: fruits may taste "sweet" (i.e. not sour) if pH is relatively high OR if TA is low. In case of acidic juice (i.e. low pH) and low TA (which means low buffering capacity) the pH of the juice will be quickly neutralized in the mouth by saliva, and thus will not taste sour.

The second paragraph of the discussion in the original manuscript, was clearly too condensed to properly to convey the relation between acidity (=pH), titratable acid and sour taste, and also confused reviewer 2 (see his/her point 1). As outlined in our reply to point 1 of reviewer 2 in more detail, we added some extra sentences in the introduction to explain that:

- sour taste is perceived when taste receptors sense high concentrations of free protons and that that requires that juice has a low pH and a high buffering capacity (Lines 31-33).
- citrate is thought to be imported into the vacuole as a conjugate base (citrate³⁻) and consequently does not lower the vacuolar pH (in fact it would increase vacuolar pH) but does increase buffering capacity (line 37-39)

In the Discussion we added a new paragraph to explain that CitPH1 and CitPH5 are directly involved in acidification of the vacuole and, indirectly, by generating the transmembrane pH gradient that is needed for the import of citrate (and other compounds) by a secondary transporters (lines 463-471).

Such as model is consistent with previous biochemical data and predicts that fruits with low CitPH1 and CITPH5 expression will all have high pH and low TA, which is indeed what we found. It also predicts that there might be mutants like 'Amber' sweet in which CitPH1/PH5 activity is normal (hence low pH), while import of citrate (and possibly other acids) is impaired, resulting in only as small amount titratable acids, low buffer capacity and eventually a sweet (non acid) taste.

Reviewer #4 (Remarks to the Author):

This paper relates to the contribution of two proton transporter in the acidification of fruit of the Citrus family, and to the identification of the CsAN1 transcription factor as the mutated locus controlling the mutant non-sour citrus genotypes studied. The research was carried out by one of the leading research groups studying proton transporters in plants, and they have been joined in this report by one of the leading citrus geneticists with a novel collection of citrus germplasm.

The research has taken a strictly genetic approach, combining a unique collection of Citrus accessions and novel mutants representing the range of citrus fruit vesicle acidification, with extensive sequencing and expression analyses.

The most significant and strongest conclusion of the research that that the loss of expression of CitPH1 and CitPH5 in the sweet mutants is correlated with the reduction in acidity, and that the mutation is in the CitAN1 transcription factor which regulates their expression. The authors show this most conclusively in their coexpression reporter gene studies in petunia protoplasts. Regretfully, research on fruit trees does not allow for the decisive in planta functional expression and complementation studies carried out on annual fruiting plants. Considering the limitations of the perennial tree system with its juvenility requirements one could not expect more.

This is an extremely important contribution to understanding the genetic physiology behind these sweet mutants and points to the contributory role of these transporters in citrus fruit acidity. The special novelty of this, as the

authors discuss, is that citrus fruit and petunia flowers share a tissue specific module of TF: functional gene interactions responsible for acidification, AN1:PH1,PH5.

The researchers expand their observations to other Citrus species, including additional lemons, oranges and pummelo/grapefruits, with novel ranges of acidity. Here too, the contribution of PH1,PH5 expression is crucial to acidification, and sometimes, but not always, related to AN1 expression levels. With this expansion of genotypes to the survey the authors conclusively show that PH1 and PH5 contribute to fruit acidification, most likely across the Citrus family, and that the regulation of these two transporters is likely under different molecular genetic control within the family. Such a range of genetic control mechanisms of a phenotype is to be expected for such downstream functional gene expression.

As such, I would recommend that this paper be published in Nature Communications as it makes an important contribution to our understanding of a subject of broad interest: plant evolution of consumable fruit leading to taste differences. However, I suggest that the paper should be revised taking into consideration the comments below.

There are some major comments, followed by minor comments.

Major comments:

1- As indicated by my positive comments, the paper shows that PH1 and PH5 expression loss is probably causal to the low acid levels in the sweet mutations and that these transporters are likely the major contributors to citrus fruit acidity across the family, including in the non-mutant accessions. However, this does not indicate that quantitative differences in expression of these two genes are causal to the broad genetic variability of acidity among the non-mutant accessions. The conclusions regarding the contribution of PH1 and PH5 expression levels to the range in fruit acidity among non-mutant citrus fruit should be limited for the following reasons:

1- Results of "relative expression" from qRT-PCR from different species, especially as normalized against actin, should be viewed with some apprehension. This is particularly so for relative values of half or even 2-fold. Without absolute read values of NGS with rigorous statistics and replications the "relative expression" leaves much to be desired and differences should be interpreted cautiously. This is especially so when comparing different accessions, and distantly related (or unrelated) lines.

REPLY: Previous microarray analyses identified several genes *CitANKYRIN*, *CitANNEXIN2* and *CitRIBOSOMAL PROTEIN S10* that are the most constantly expressed in broad range of tissues from different Citrus species (see Aprile et al 2011; ref 33). We performed additional qRT-PCR experiments in which we measured the RNAs of the three aforementioned genes in all the RNA samples used for the data in Figs 1, 3, 5 and 6 and Supplemental Fig 1 and recalculated relative expression levels using now 4 reference genes (Actin, and the three above mentioned genes). These data are now shown in the revised versions of Figs 1,3, 5 and 6 and Supplemental Fig 1. Note that this did not alter the result or their interpretation (although it does make the data more robust).

We also emphasize that our conclusions rest of large (near "all or nothing") changes in mRNA levels. Given that the Citrus varieties are not isogenic and were not grown under identical circumstances a myriad of factors (genetic or environmental) may cause small variations in cellular parameters (pH, TA etc) and mRNA levels. Therefore, we made no attempts to relate small differences between various acidic (non-mutant) varieties to differences in CitPH1 or CitPH5 expression. In fact, we added a new paragraph in the discussion where this is explicitly pointed out this (lines 455-462).

2- Even the harvest stage chosen can be a confounding factor as each accession likely has a somewhat different developmental pattern of fruit development (in fact, some of the differences in acidity between lines may in fact

be related to different developmental stages at harvest). It is well known that citrus fruit acidity undergoes major developmental fluctuations. The authors do not describe in the methods section the stage of fruit development (this should be done as descriptively as possible- realizing that flowers were not tagged at anthesis) but it is unlikely, or even impossible, that there is any consistency in fruit stages. In an ideal situation each line would be studied under a number of defined developmental stages in order to extrapolate the contribution of expression levels to acid accumulation. But without this the authors should be conservative with their conclusions.

REPLY: We added a morphological description of the developmental stages of the fruits that were used in the legends of Figs 1, 3, 5 and 6. We are well aware that aside for differences in developmental stage, many other factors may cause (relatively) small decreases or increases in expression levels (see our reply to comment 3 below).

3- A further indication that the expression levels (at only a single developmental stage) of PH1 and PH5 do not explain fruit acidification across the citrus germplasm is the difference in acidification between lemons and the other acidic non-mutant citrus. Lemons undergo a larger "hyperacidification than do sour oranges and from the qRTPCR results the expression of PH1 and PH5 at a single developmental stage cannot explain the difference. In fact, a simple correlation between acidity (pH?) and expression of PH1 and PH5 across the non-mutant accessions shows that there is little correlation. This indicates that while PH1 and PH5 are indeed major genes with epistatic control over citrus acidification, their role in being causal to the genetic variability among non-mutant Citrus cannot be ascertained from this study and indications from this study are in fact that there may be little (e.g., PH1 and PH5 expression levels among the different sour orange varieties in Fig. 6).

In conclusion, the major comment is that the paper should be somewhat modified taking the above points into consideration. The clear conclusions relating to the role of these transporters and the mutation in AN in the mutant lines should be separated from the broader picture of acidification in the non-mutant Citrus lines. For the latter, the authors may conclude that the two transporters do indeed contribute but that it cannot be concluded that genetic variability for them contributes to the variability in acid levels within the non-mutant family. Also, it may not be cautious to claim that (line 237) PH1, PH5 are "responsible for hyperacidification" and the term perhaps should be preserved for the extreme acidic sour lemons, as was by Muller et al (1996).

REPLY:

In the paper we only drew conclusions from the large –nearly "all or nothing" – differences in PH1/PH5 expression between sour (low-pH) and sweet (high-pH) fruits, but we deliberately did not say much (or conclude anything) from the smaller differences in PH1 and PH5 mRNA abundance between different sour (low pH fruits), as many other factors might contribute to these smaller pH differences:

- One cannot grow citrus trees under exactly similar conditions as one can do with model plants like Arabidopsis (in growth chambers etc.). Hence, small differences in soil conditions, shadowing by neighboring trees and buildings, day by day variation in weather conditions etc. will inevitably contribute to small variations in pH. For these reasons we included in our analyses also the chimeras Faris and Amber, as these enable one to compare sweet and sour fruits that develop under identical conditions on a single tree
- As this reviewer noted correctly small variations in mRNAs levels and pH values may also originate from small differences in developmental stages of the fruits. We did our best to minimize these differences, but completely eliminating them is hardly possible.
- It is important to note that vacuolar pH is not only determined by pumps that put protons into the vacuole, but also by secondary transporters, like antiporters, that important other solutes into the vacuole in exchange for protons. In petunia, for example, overexpression of the sodium-proton exchanger NHX, reduces vacuolar acidity to the point that petals turn blue, quite similar to loss of function mutations

in a *PH* gene (see Faraco et al 2014, ref 28). Hence it is possible that variation in the expression of NHX or other proton antiporters contributes to the small pH differences between sour (low pH) fruits.

Additionally, the authors ruled out the possible contribution of the SOUR transporter to acidification in the mutant lines based on expression levels. While for the other genes studied the authors also sequenced the gene to be certain that only expression levels were of relevance, for the SOUR gene they did not sequence the different alleles. However, the SOUR gene may still be involved since it was shown that the evolution of non-sour in melon was due to a change in gene sequence rather than expression. Although it seems most likely that the AN:PH1PH5 mutation is indeed causal in the mutant citrus, nevertheless the SOUR sequence should be reported as well. Furthermore, it still remains a possibility that SOUR expression levels are related to the genetic variability in the non-mutant citrus. Although this may be out of the scope of the present paper, I imagine the authors still have all the RNA and cDNA samples and a qRT-PCR for the SOUR gene should be relatively painless and might offer some interesting insights.

REPLY: In the original manuscript we reported mRNA expression levels of the *SOUR* homolog (*CitSO*), as determined by qRT-PCR in Supplemental Fig 1. Although *CitSO* mRNA abundance does vary between the fruits of different varieties, that does not correlate with taste of the fruit or acidity of the juice.

As *SO* expression does increase acidity (lowering pH) of the fruit in melon and tomato, it is quite possible that it contributes to the acidity of *Citrus* fruits too. To compare the contribution of *SO* and PH1/PH5 to acidification of the fruit it is important to consider the following:

1. Although homologs of *CitPH1* and *CitPH5* are widespread they are missing in a few species (Li et al 2016). As melon and tomato lack both *PH1* and *PH5*, variation in *SO* expression is predicted to have a bigger impact in these species than in other species that do express *PH5* and *PH1*.
2. It is unclear *how* *SO* lowers fruits pH, since *SO* seems to localize in the ER, while fruit acidity is thought to be determined primarily by vacuolar pH.
3. In *Citrus* the down regulation of PH1/PH5 strongly reduces the accumulation of titratable acid (which in *Citrus* is mostly citric acid). Given that the translocation of citrate into the vacuole is driven by the pH gradient across the tonoplast, reduced *CitPH1/PH5* expression would be predicted to reduce citric acid accumulation. We indeed find a strong reduction of titratable acid in all sweet (high-pH) citrus fruits. Inactivation of *SO* in melon and tomato, by contrast, reduces citric acid levels by only 50% and malic acid even less (Cohen et al 2014; ref 36), supporting that the effect of *SO* on vacuolar pH is much less than that of PH1/PH5.

That said, we do agree that it would strengthen the paper to examine whether mutations occurred in *CitSO* that would lead to inactivation without a dramatic effect on *CitSO* transcript abundance. Therefore, we sequenced *CitSO* mRNAs from all analyzed varieties. The sequences, which are shown as an alignment in the new Supplementary Figure 2, did not reveal any obvious inactivating mutations in *CitSO*.

I would also suggest that the authors modify the title of the paper, "Flower color genes make citrus fruit sour". Although catchy, it is really incorrect, and seems inappropriate for a Nature Communications paper. PH1 and PH5 are not "flower color genes" but as the authors describe in their Li et al 2016 Evolution paper, they are ubiquitous tonoplast P-ATPase transporters. That may sound less sexy but the title should be a scientific description of the major point.

REPLY: We still like the original title "Flower color genes make *Citrus* fruit sour" very much, but we agree that it might be perhaps better suited for a News and Views story.

Therefore, we changed the title into: Hyperacidification of *Citrus* fruits by a vacuolar proton-pumping P-ATPase

complex

Minor comments:

1- "brix" should be presented as degrees (Brix0), not %, and does not represent the soluble sugar content but rather, as an indication of refraction, all soluble solids. Sugars are the major soluble solids that accumulate and change with development and as such Brix can parallel developmental changes in sugar levels. But since it is not actually sugar the correlation between Brix and sugar is not strong enough to extrapolate that a difference in Brix of 1, for example, is related to a parallel difference in sugar.

REPLY: We agree and adjust the ms as follows:

- we now present Brix as degrees instead of percentage in all Figures
- in the text we replaced "sugar content (Brix)" with "soluble solid content (Brix)"
- On first use (line 100), we added that sugars are the major constituents of the soluble solids.

2- Perhaps my copy was printed poorly but my Fig. 4 for Sweet Lemon shows one allele with the central 4 exons in light grey, indicating a splice variant, and the second allele is all grey, indicating silencing (?). The results section (line 194) describes otherwise.

REPLY: We wanted to indicate for the shown alleles which structures were determined by complete sequencing and which structures were inferred from PCR data. We had specified in the legend to Fig 4 that "Gene structures inferred from sequencing are marked by dark colors; structure inferred from PCR alone are marked by a weaker coloration (reduced saturation). Possibly the difference dark and weak coloration was not clear enough and too much dependent on the quality of prints. Hence, we revised the design of Fig 4 as follows:

- Fully sequenced exons are indicated with a dark grey filling, and sequenced intron and 5' and 3' flanking regions with a dark grey line.
- Exons that were inferred from PCR data only are now indicated by a solid grey line and white filling (instead of light gray), which should make them better distinguishable from the sequenced dark grey exons, even when the quality of the print is suboptimal.
- We now marked deletions with a red dotted line (instead of light blue), making them stand out better from other elements. If both breakpoints were confirmed by sequencing, the deletion is indicated with pink filling, if the deletion was inferred from PCR data alone by white filling.

3- In Fig. 3d, for the expression of AN1 in Sweet Lemon, which 3' primers were used? Did they distinguish between the deletions?

REPLY: As outlined above in our reply to point 5 of reviewer 2, the qRT-PCR data shown in the paper we generated 1st strand cDNA and performed q-PCR assays of the different RNA samples all side by side in one big experiments/measurements. CitAN1 mRNA was measured in all samples with 3 distinct primer pairs amplifying a region in the 5' end (from exon 1 to exon 2, across intron 1), a region in the middle and a region in the 3' end. For 'Faris' sweet, 'Faris' sour and, as a control 'Frost Lisbon', we show the result for all three primer pairs, because it demonstrates that 'Faris' sweet *citn1* mRNA lacks the 3' region.

For the varieties in Figs 3, 5 and 6 all three primer pairs gave essentially similar results (as expected, because none of these varieties are homozygous for a deletion). To avoid redundancy we have shown only the results obtained with the most 5' primer pair.

Hence the data shown on *CitAN1* RNA from Sweet lemon are with a primer pair that would have detected expression from both alleles. The other primer pairs gave the same result, and therefore not shown.

4- In Fig. 1f,mutant- Indicate what the different dotted lines leading to the 3' end represent in the legend.

REPLY: As mentioned above in our reply to comment #2 above, we revised the color code in Fig 4. We adjusted the legend accordingly.

In conclusion, I commend the authors on a very significant paper but suggest the following revisions

1- Modification of the narrative and discussion, emphasizing that the research results indicate that the non-sour mutations point to the major role of PH1 and PH5 in citrus acidification but do not imply that they are responsible for the broad genetic variability of acidification in the family.

2- Description of the sampling stages and referral to the limitations of a single harvest stage in such a study.

3- Sequences of the SOUR alleles and expression of SOUR in the other citrus genotypes.

4- Modification of the title to be more descriptive.

5- The 4 minor comments.

REPLY: these points have all been dealt with above.

REVIEWERS' COMMENTS:

Reviewer #2 (Remarks to the Author):

I think that this is an improved version of the manuscript and the authors did a nice job addressing the questions I had in the previous version.

A minor detail, is related to this sentence in the discussion that probably can be improved. "We have shown that juice vesicles cells of all Citrus varieties with acidic (low pH) fruits express CitPH1 and CitPH5, encoding a vacuolar P ATPase proton pump, while in acidless Citrus fruit varieties in which acidity of the fruit strongly reduced, CitPH1 and CitPH5 expression levels are drastically reduced".

Reviewer #4 (Remarks to the Author):

The revised manuscript is a significant improvement of the original version and addresses all the questions raised in the first review. I am happy to recommend it for publication in Nature Communications.

RESPONSE TO REVIEWERS' COMMENTS:

Reviewer #2 (Remarks to the Author):

I think that this is an improved version of the manuscript and the authors did a nice job addressing the questions I had in the previous version.

- Thank you

A minor detail, is related to this sentence in the discussion that probably can be improved.

“We have shown that juice vesicles cells of all Citrus varieties with acidic (low pH) fruits express CitPH1 and CitPH5, encoding a vacuolar P -ATPase proton pump, while in acidless Citrus fruit varieties in which acidity of the fruit strongly reduced, CitPH1 and CitPH5 expression levels are drastically reduced”.

- We rephrased that sentence as follows: “We have shown that juice vesicles of *Citrus* varieties with acidic (low pH) fruits express CitPH1 and CitPH5, encoding two interacting P -ATPases that constitute a vacuolar proton pump, while, CitPH1 and CitPH5 expression levels are drastically decreased in fruit varieties with reduced acidity (high pH).”

Reviewer #4 (Remarks to the Author):

The revised manuscript is a significant improvement of the original version and addresses all the questions raised in the first review. I am happy to recommend it for publication in Nature Communications.

- Thank you.